# Who is Afraid of Big Bad Minima? Analysis of Gradient-Flow in a Spiked Matrix-Tensor Model

**Stefano Sarao Mannelli**[†], **Giulio Biroli**[‡], **Chiara Cammarota**[∗],
**Florent Krzakala**[‡], **and Lenka Zdeborová**[†]

## Abstract

Gradient-based algorithms are effective for many machine learning tasks, but despite ample recent effort and some progress, it often remains unclear why they work in practice in optimising high-dimensional non-convex functions and why they find good minima instead of being trapped in spurious ones. Here we present a quantitative theory explaining this behaviour in a spiked matrix-tensor model. Our framework is based on the Kac-Rice analysis of stationary points and a closed-form analysis of gradient-flow originating from statistical physics. We show that there is a well defined region of parameters where the gradient-flow algorithm finds a good global minimum despite the presence of exponentially many spurious local minima. We show that this is achieved by surfing on saddles that have strong negative direction towards the global minima, a phenomenon that is connected to a BBP-type threshold in the Hessian describing the critical points of the landscapes.

## 1   Introduction

A common theme in machine learning and optimisation is to understand the behaviour of gradient descent methods for non-convex problems with many minima. Despite the non-convexity, such methods often successfully optimise models such as neural networks, matrix completion and tensor factorisation. This has motivated a recent spur in research attempting to characterise the properties of the loss landscape that may shed some light on the reason of such success. Without the aim of being exhaustive these include [1, 2, 3, 4, 5, 6, 7, 8, 9, 10, 11].

Over the last few years, a popular line of research has shown, for a variety of systems, that spurious local minima are not present in certain regimes of parameters. When the signal-to-noise ratio is large enough, the success of gradient descent can thus be understood by a trivialisation transition in the loss landscape: either there is only a single minima, or all minima become "good", and no spurious minima can trap the dynamics. This is what happens, for instance, in the limit of small noise and abundance of data for matrix completion and tensor factorization [3, 8], or for some very large neural networks [1, 2]. However, it is often observed in practice that these guarantees fall short of explaining the success of gradient descent, that is empirically observed to find good minima very far from the regime under mathematical control. In fact, gradient-descent-based algorithms may be able to perform well even when spurious local minima are present because the basins of attraction of the spurious minima may be small and the dynamics might be able to avoid them. Understanding this behaviour requires, however, a very detailed characterisation of the dynamics and of the landscape, a feat which is not yet possible in full generality.

∗ Department of Mathematics, King's College London, Strand London WC2R 2LS, UK.

A fruitful direction is the study of Gaussian functions on the $N$-dimensional sphere, known as $p$-spin spherical spin glass models in the physics literature, and as isotropic models in the Gaussian process literature [12, 13, 14, 15, 16]. In statistics and machine learning, these models have appeared following the studies of spiked matrix and tensor models [17, 18, 19, 20, 21]. In particular, a very recent work [22] showed explicitly that for a spiked matrix-tensor model the gradient-flow algorithm indeed reaches global minimum even when spurious local minima are present and the authors estimated numerically the corresponding regions of parameters. In this work we consider this very same model and explain the mechanism by which the spurious local minima are avoided, and develop a quantitative theoretical framework that we believe has a strong potential to be generic and extendable to a much broader range of models in high-dimensional inference and neural networks.

**The Spiked Matrix-Tensor Model**. The spiked matrix-tensor model has been recently proposed to be a prototypical model for non-convex high-dimensional optimisation where several non-trivial regimes of cost-landscapes can be displayed quantitatively by tuning the parameters [23, 22]. Related mixed matrix-tensor models have also been studied in the context of text-analysis applications in [20, 19]. In this model, one aims at reconstructing a hidden vector (i.e. the spike) $\boldsymbol{\sigma}^*$ from the observation of a noisy version of *both* the rank-one matrix and rank-one tensor created from the spike. Using the following notation: bold lowercase symbols represent vectors, bold uppercase symbols represent matrices or tensors, and $\langle \cdot, \cdot \rangle$ represent the scalar product, the model is defined as follows: given a signal (or spike), $\boldsymbol{\sigma}^*$, uniformly sampled on the $N$-dimensional hyper-sphere of radius 1, it is given a tensor $\boldsymbol{T}$ and a matrix $\boldsymbol{Y}$ such that

$$T_{i_1 \ldots i_p} = \eta_{i_1 \ldots i_p} + \sqrt{N(p-1)!}\, \sigma_{i_1}^* \ldots \sigma_{i_p}^*, \tag{1}$$

$$Y_{ij} = \eta_{ij} + \sqrt{N}\sigma_i^* \sigma_j^*, \tag{2}$$

where $\eta_{i_1 \ldots i_p}$ and $\eta_{ij}$ are Gaussian random variables of variance $\Delta_p$ and $\Delta_2$ respectively. Neglecting constant terms, the maximum likelihood estimation of the ground truth, $\boldsymbol{\sigma}^*$, corresponds to minimization of the following loss function:

$$\ell(\boldsymbol{\sigma}|\boldsymbol{T}, \boldsymbol{Y}) = -\frac{\sqrt{(p-1)!}}{\Delta_p \sqrt{N}} \sum_{i_1 < \cdots < i_p} T_{i_1 \ldots i_p} \sigma_{i_1} \ldots \sigma_{i_p} - \frac{1}{\Delta_2 \sqrt{N}} \sum_{i<j} Y_{ij} \sigma_i \sigma_j = \tag{3}$$

$$-\frac{\sqrt{(p-1)!}}{\Delta_p \sqrt{N}} \sum_{i_1 < \cdots < i_p} \eta_{i_1 \ldots i_p} \sigma_{i_1} \ldots \sigma_{i_p} - \frac{1}{\Delta_2 \sqrt{N}} \sum_{i<j} \eta_{ij} \sigma_i \sigma_j - \frac{\langle \boldsymbol{\sigma}, \boldsymbol{\sigma}^* \rangle^p}{p \Delta_p} - \frac{\langle \boldsymbol{\sigma}, \boldsymbol{\sigma}^* \rangle^2}{2 \Delta_2}.$$

The first two contributions of the last equation will be denoted $\epsilon_p$ and $\epsilon_2$ in the following. While the matricial observations correspond to a quadratic term and thus to a simple loss-landscape, the additional order-$p$ tensor contributes towards a rough non-convex loss landscape. As $\Delta_p \to \infty$ the information $T_{i_1 \ldots i_p}$ becomes irrelevant and the landscape becomes trivial, while in the opposite limit $\Delta_2 \to \infty$, the landscape becomes extremely rough and complex as analyzed recently in [16, 24].

We shall consider the behaviour of the gradient-flow (GF) algorithm aiming to minimise the loss:

$$\dot{\sigma}_i(t) = -\mu(t)\sigma_i(t) - \frac{\partial \ell(\boldsymbol{\sigma}(t)|\boldsymbol{T}, \boldsymbol{Y})}{\partial \sigma_i(t)}, \tag{4}$$

where $\mu(t)$ enforces that $\boldsymbol{\sigma}(t)$ belongs to the hyper-sphere of radius $N$ and will be referred to as the *spherical constraint*. The algorithm is initialised in a random point drawn uniformly on the hyper-sphere, thus initially having no correlation with the ground-truth signal. We view the gradient-flow as a prototype of gradient-descent-based algorithms that are the work-horse of current machine learning.

**Main Contributions**. The first main result of this paper is the expression for the threshold below which the gradient-flow algorithm finds a configuration correlated with the hidden spike. This threshold is established in the asymptotic limit of large $N$, fixed $p$ and $\Delta_p$, and reads:

$$\Delta_2^{\mathrm{GF}}(\Delta_p, p) \equiv \frac{-\Delta_p + \sqrt{\Delta_p^2 + 4(p-1)\Delta_p}}{2(p-1)}. \tag{5}$$

We find that (i) for $\Delta_2 < \Delta_2^{\mathrm{GF}}$ the gradient flow reaches in finite time the global minimum, well correlated with the signal, while (ii) for $\Delta_2 > \Delta_2^{\mathrm{GF}}$ the algorithm remains uncorrelated with the signal for all times that do not grow as $N$ grows. We contrast it with the threshold $\Delta_2^{\mathrm{triv}} < \Delta_2^{\mathrm{GF}}$,

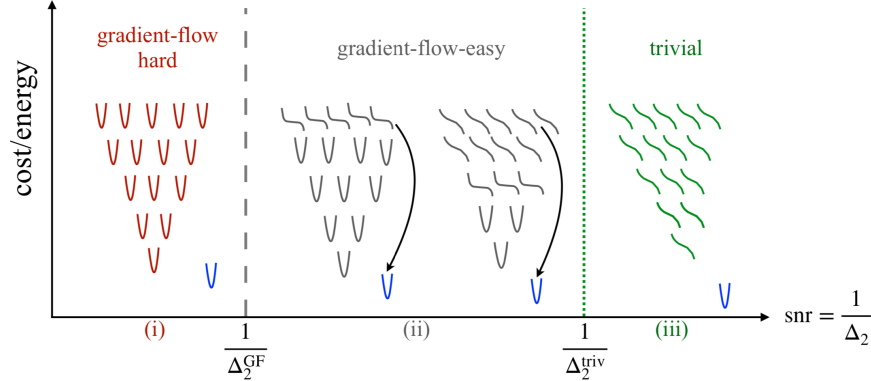

Figure 1: Cartoon illustrating the mechanism by which gradient-flow avoids spurious local minima in the spiked matrix-tensor model. As the signal-to-noise (snr) ratio $1/\Delta_2$ is increased, the spurious local minima that attract the randomly initialised GF algorithm develop a single negative direction towards the global minimum before the others, in particular the lower-cost spurious local minima, do. This has drastic consequences on the GF algorithm. In region (i), for $\text{snr} < 1/\Delta_2^{\text{GF}}$, the algorithm goes down the landscape, eventually reaches the high-energy *threshold* minima and remains stuck. In region (ii), however, these threshold minima are turned into saddles with a strong negative direction towards the signal. The algorithm is initially reaching these minima-turned-saddles, surfing on the negative slope, it then turns towards the "good" minima correlated with the signal, avoiding the exponentially many spurious minima at lower energies. The main technical contribution of this paper is a quantitative description of this scenario, including a simple formula for the corresponding threshold $\Delta_2^{\text{GF}}$, eq. (5). As the snr is further increased, the negative direction appears in lower and lower minima until the trivialization transition in region (iii): for $\text{snr} > 1/\Delta_2^{\text{triv}}$, all the spurious minima have been turned into saddles.

established in [22], below which the energy landscape does not present any spurious local minima. Note that $\Delta_2^{\text{GF}}$ is less than $\Delta_2^{\text{AMP}} = 1$ [23], below which the best known algorithm, specifically the approximate message passing, works.

The second main result of this paper, is the insight we obtain on the behaviour of the gradient-flow in the loss landscape, that is summarised in Fig. 1. The key point is to consider the fate of the spurious local minima that attract the GF algorithm when the signal to noise ratio $\text{snr} = 1/\Delta_2$ is increased. As the snr increases, these minima turn into saddles with a single negative direction towards the signal (a phenomenon that we analyze in the next section, and that turns out to be linked to the BBP transition [25] in random matrix theory), all that well before all the other spurious local minima disappear. We present two ways to quantify this insight:

(a) We use the Kac-Rice formula for the number of stationary points, as derived for the present model in [22]. In [22] this formula is used to quantify the region with no spurious local minima. Here we focus on a BBP-type of phase transition that is crucial in the derivation of this formula and deduce the GF threshold (5) from it.

(b) We use the CHSCK equations [26, 27] for closed-form description of the behaviour of the gradient-flow, as derived and numerically solved in [23, 22]. Building on dynamical theory of mean-field spin glasses we determine precisely when and how the algorithm escapes the manifold of zero overlap with the signal, leading again to the threshold (5).

Both these arguments are derived using reasoning common in theoretical physics. From a mathematically rigorous point of view the threshold (5) remains a conjecture and its rigorous proof is an interesting challenge for future work. We note that both the Kac-Rice approach [16] and the CHSCK equations [28] have been made rigorous in closely related problems.

Moreover, our reasoning leading to the formula for $\Delta_2^{\text{GF}}$ is in retrospect not restricted to the present model, and ends up way simpler than the full CHSCK analysis or the full Kac-Rice complexity calculation. The mechanism of converging to the threshold states and then escaping from them can also be tested numerically even in models that are not amenable to analytic description. This makes the results of the present work widely testable and applicable to other settings than the present model.

Therefore, we will investigate analytically and numerically other models and real-data-based learning in order to validate this theory and to understand its limitations.

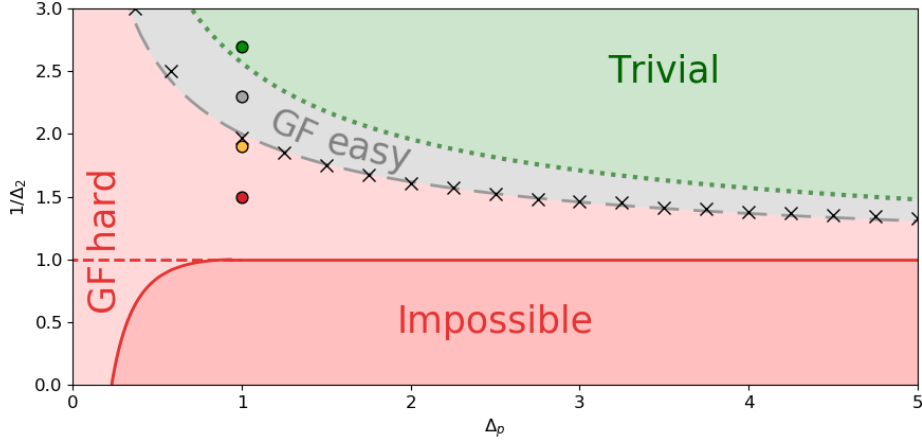

Figure 2: The phase diagram shows the different regions for gradient-flow behaviour, for the spiked matrix-tensor model with $p = 3$. In the region shaded in red (light and dark), GF does not correlate with the signal, while it does in the grey and green regions. In the dark-red region obtaining correlation with the signal is *impossible* information-theoretically [23]. The possible region is divided by a red dashed line, below that line even best known algorithms are unable to obtain correlation with the signal [23]. The green region is characterised by a *trivial* landscape, i.e. all the spurious minima disappear [22]. The grey region is where gradient-flow succeeds to converge despite the presence of spurious minima. We marked with black crosses points predicting the gradient-flow threshold obtained numerically in [22], they perfectly agree with our theoretical prediction of the threshold (5), marked by the grey dashed line. The circles in colours are points that we will use to illustrate the different features of these regions.

## 2 Probing the Landscape by the Kac-Rice Method

The statistical properties of the landscape associated to the loss function (3) can be studied by the Kac-Rice method. The technique was developed in the 40s in [29, 30], nicely summarized in [31], and applied in high-dimensional problems in [13, 14]. Earlier applications in high dimensions appear in the statistical physics literature, see [32] for an overview, and has been recently extended in [24].

The quantities of interest are the number of critical points at a given energy, $\mathcal{N}(\epsilon_p, \epsilon_2)$, and the Hessian matrix evaluated at those critical points. We analyse the logarithm of $\mathcal{N}(\epsilon_p, \epsilon_2)$, called the *complexity*. Since the complexity is a random quantity we compute its upper bound $\Sigma_a(\epsilon_p, \epsilon_2) = \ln \mathbb{E}[\mathcal{N}(\epsilon_p, \epsilon_2)]$, along the lines of [16, 22]. We have also computed its typical value $\Sigma_q(\epsilon_p, \epsilon_2) = \mathbb{E}[\ln \mathcal{N}(\epsilon_p, \epsilon_2)]$ along the lines of [24], i.e. non-rigorously using the replica symmetry assumption (see SM Sec. A). In what follows we focus on complexity of stationary points with no correlation with the signal, in which case analytical and numerical arguments (see SM Sec. A.2.1) indicate that $\Sigma_a(e_p, e_2)$ and $\Sigma_q(e_p, e_2)$ are either very close numerically or possibly equal, and by Jansen inequality it implies a bound on its asymptotic distribution. Thus, in the following, we will simply refer to the complexity $\Sigma(\epsilon_p, \epsilon_2)$ without further specification.

In the Kac-Rice analysis the statistics of the Hessian, $\mathcal{H}$, of critical points plays a key role. It was shown in [22], and the argumentation is reproduced in the SM Sec. A.2, that $\mathcal{H}$ has a simple form for the loss (3). It is a $(N-1) \times (N-1)$ matrix formed by the sum of three contributions: a random matrix $\mathbb{W}_{N-1}$ belonging to the Gaussian orthogonal ensemble (GOE), a matrix proportional to the identity, and a rank one projector in the direction of the signal. Under the choice of a convenient reference frame [22] where the first element of the basis $e_1$ is aligned with the components of the estimator tangent to the signal, the expression of $\mathcal{H}$ for critical points with null overlap $m$ with the

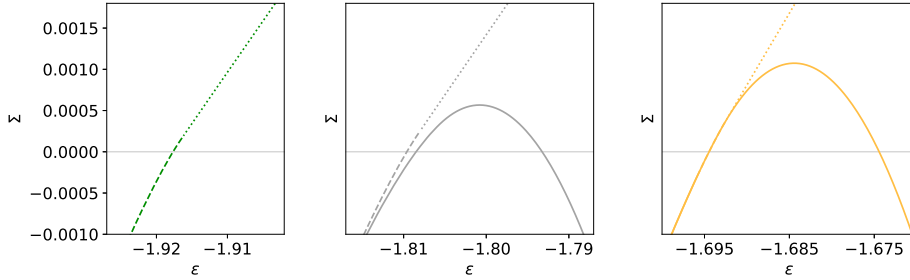

Figure 3: Complexity curves for the number of critical points for an overlap value $m = 0$ at fixed $\Delta_p = 1.0$ for (from left to right) $1/\Delta_2 = 2.7, 2.3, 1.9$. The lines are dotted when the complexity is dominate by critical points having an extensive number of eigenvalues are negative, dashed when only one eigenvalue is negative, full when the points have only positive (or null) eigenvalues i.e. they are minima. The complexity of the minima is drawn in full lines with the same colours, and it merges with the complexity of stationary points when it becomes dominant.

signal and with energies $\epsilon_p$ and $\epsilon_2$ reads:

$$\mathcal{H} = \sqrt{Q''(1)} \left[ \mathbb{W}_{N-1} + t\, \mathbb{I}_{N-1} - \theta\, \boldsymbol{e}_1 \boldsymbol{e}_1^T \right] \tag{6}$$

with $Q(x) = \frac{x^p}{p\Delta_p} + \frac{x^2}{2\Delta_2}$, $t = -\left(p\epsilon_p + 2\epsilon_2\right)/\sqrt{Q''(1)}$, and $\theta = Q''(0)/\sqrt{Q''(1)}$. The normalisation of $\mathbb{W}_{N-1}$ is chosen such that $\mathrm{Tr}\mathbb{E}[\mathbb{W}_{N-1}^2] = 1$.

**The Fate of the Spurious:** The initial condition for the gradient-flow algorithm is a random configuration $\boldsymbol{\sigma}_0$ uniformly drawn on the hyper-sphere. Such an initial condition clearly belongs to the large manifold of configurations uncorrelated with the ground-truth signal. We aim to investigate how does the gradient flow manage to escape from this initial manifold. For this purpose we focus on the properties of the landscape in the subspace where the overlap with the signal is zero, $m = 0$.

In Fig. 3, we plot the complexity at $m = 0$ as a function of the energy $\epsilon$

$$\Sigma(\epsilon) = \sup_{\substack{\epsilon_p, \epsilon_2 \\ \text{s.t. } \epsilon_p + \epsilon_2 = \epsilon}} \Sigma(\epsilon_p, \epsilon_2) \Big|_{m=0}$$

for the points $1/\Delta_2 = 1.9, 2.3, 2.7$ and $\Delta_p = 1.0$ ($p = 3$), which are marked with circles of the corresponding colour in Fig. 2. We use discontinuous lines for the complexity of critical points that have at least one negative direction, and full lines for the complexity of local minima. A finding of [22], that holds for any value of $\Delta_p$, is that for small $1/\Delta_2$ the majority of critical points with zero overlap with the signal at low enough energies are spurious minima; they disappear increasing $1/\Delta_2$ above a $\Delta_p$-dependent value $1/\Delta_2^{\text{triv}}$ corresponding to the green region of Fig. 2. In this part of the phase diagram, there are no spurious minima and the global minimum is correlated with the signal; this is an "easy" landscape for gradient flow which is therefore expected to succeed there. The main open question concerns the behavior for smaller values of $1/\Delta_2$: When does the existence of spurious minima, appearing in panel (b) and (c) of Fig. 3, start to be harmful to gradient flow?

In order to answer this question, we investigate more closely the nature of the spurious minima at different energies. We focus in particular on their Hessian, which plays a crucial role in order to understand which spurious minima have the largest basin of attraction and, hence, can trap the algorithm. Specifically, the Hessian Eq. (6) at a critical point is the sum of a GOE matrix, a multiple $t$ of the identity and rank one perturbation with strength $\theta(\Delta_2, \Delta_p)$. For low signal-to-noise ratio, large $\Delta_p$ and large $\Delta_2$, the spectrum of (6) is a shifted Wigner semicircle with support $\left[\sqrt{Q''(1)}(-2 + t), \sqrt{Q''(1)}(2 + t)\right]$. The most numerous critical points at fixed energy $\epsilon$ are characterized by a $t(\epsilon)$ that is a monotonously decreasing function of $\epsilon$, Fig. A.2 in the SM. Moving towards higher energies, the spectrum of the Hessian shifts to the left, which indicates smaller curvature and wider minima. The transition between minima and saddles takes place at the *threshold energy* when the left edge of the Wigner semi-circle law touches zero, i.e. when $t(\epsilon_{\text{th}}) = 2$, the numerical value is obtained in the Appendix Sec. A.2.3. Putting the above findings together, minima at $\epsilon = \epsilon_{\text{th}}$ are the most numerous and *the marginally stable ones*. Therefore, they are the

natural candidates for having the largest basin of attraction and the highest influence on the randomly initialised algorithm. This reasonable guess is at the basis of the theory of glassy dynamics in physics [27]. We take it as a working hypothesis for now, and we confirm it analytically and numerically in what follows. We also remark that this phenomenology can be tested also in models that, contrarily to the present one, are not amenable to an analytic description.

Finally, the effect of the perturbation (third contribution to the RHS of (6)) on the support of the spectrum is negligible as long as $\theta \leq 1$, as follows from the work on low-rank perturbations of random GOE matrices [25]. Thus, when the signal-to-noise ratio is small we expect that the configuration $\boldsymbol{\sigma}(t)$ slowly approaches at long times the ubiquitous "threshold minima" characterised by energy $\epsilon_{\text{th}}$ and zero overlap with the signal.

The last missing piece is unveiling what makes those minima unstable for large snr. We show below that it is a transition, called BBP (Baik-Ben Arous-Péché) [25], which takes place in the spectrum of the Hessian when $\Delta_2$ and $\Delta_p$ become small enough so that $\theta$ becomes larger than one, as in Fig. 3. After the transition an eigenvalue, equal to $\sqrt{Q''(1)}\left(-\theta - \theta^{-1} + t\right)$, pops out on the left of the Wigner semi-circle, and its corresponding eigenvector develops a finite overlap with the signal [25]. This implies that, as soon as the isolated eigenvalue pops out, an unstable downward direction towards the signal emerges in the landscape around the threshold minima, at which point the algorithmic threshold for gradient flow takes place. Interestingly, many other spurious minima at lower energy also undergo the BBP transition, but they remain stable for longer as the isolated eigenvalue is positive when it pops out from the semi-circle. In conclusion, our analysis of the landscape suggests a dynamical transition for signal estimation by gradient flow given by

$$\theta = Q''(0)/\sqrt{Q''(1)} = 1 \tag{7}$$

which leads to a very simple expression for the transition line $\Delta_2^{\text{GF}}$, Eq. (5). This theoretical prediction is shown in Fig. 2 as a dashed grey line: The agreement with the numerical estimation from [22] (black crosses) is perfect.

Our analysis unveils that the key property of the loss-landscape determining the performance of the gradient-flow algorithm, is the (in)stability in the direction of the signal of the minima with largest basin of attraction. These are the most numerous and the highest in energy, a condition that likely holds for many high-dimensional estimation problems.

The other spurious minima, which are potentially more trapping than the threshold ones and still stable at the algorithmic transition just derived, are actually completely innocuous since a random initial condition does not lie in their basin of attraction with probability one in the large $N$ limit. This benign role of very bad spurious minima might appear surprising; it is due to the high-dimensionality of the non-convex loss function. Indeed it does not happen in finite dimensional cases, in which a random initial condition has instead a finite probability to fall into bad minima if those are present.

## 3 Probing the Gradient-Flow Dynamics

### 3.1 Closed-Form Dynamical Equations

In the large $N$ limit gradient-flow dynamics for the spiked matrix-tensor model can be analysed using techniques originally developed in statistical physics studies of spin-glasses [33, 34, 35] and later put on a rigorous basis in [28]. Three observables play a key role in this theory:

(i) The overlap (or correlation) of the estimator at two different times: $C(t, t') = \langle \boldsymbol{\sigma}(t), \boldsymbol{\sigma}(t') \rangle$.

(ii) The change (or response) of the estimator at time $t$ due to an infinitesimal perturbation in the loss at time $t'$, i.e. $\ell \rightarrow \ell + \langle \boldsymbol{\sigma}(t'), \boldsymbol{h}(t') \rangle$ in Eq. (4): $R(t, t') = \sum_{i=1}^{N} \frac{\delta \sigma_i(t)}{\delta h_i(t')}\Big|_{h_i=0}$.

(iii) The average overlap of the estimator with the ground truth $m(t) = \langle \boldsymbol{\sigma}^*, \boldsymbol{\sigma}(t) \rangle$.

For $N \rightarrow \infty$ the above quantities converge to a non-fluctuating limit, i.e. they concentrate with respect to the randomness in the initial condition and in the generative process, and satisfy closed equations. Following works of Crisanti-Horner-Sommers-Cugliandolo-Kurchan (CHSCK) [34, 35]

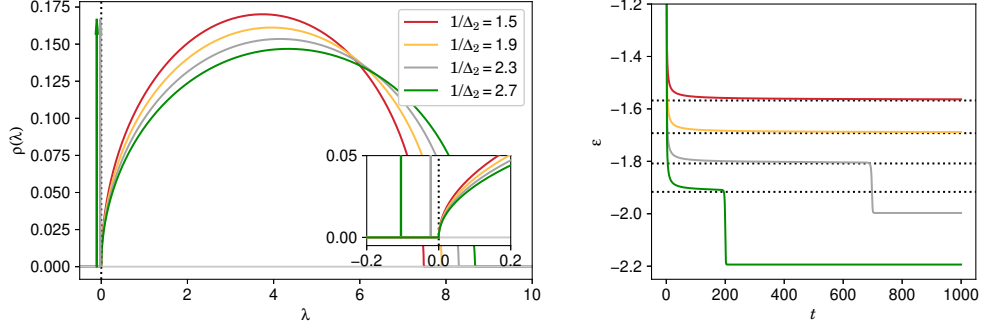

Figure 4: Right panel: energy as a function of time for the set of parameters indicated by small circles in Fig. 2. The horizontal dotted lines correspond to value of the threshold energy $\epsilon_{\text{th}}$, as derived both from the Kac-Rice approach in Appendix Sec. A.2.3 and from the large time behaviour of the dynamics in Appendix Sec. B.2.6. Left panel: Eigenvalue distribution of the Hessian of the threshold states for the same set of parameters. When $1/\Delta_2$ becomes smaller than 2 an isolated eigenvalue appears; it has been highlighted using vertical arrows. Concomitantly, the energy as a function of time first approaches the plateau and eventually departs from it and reaches the energy of the global minimum.

and their recent extension to the spiked matrix-tensor model [23, 22] the above quantities satisfy:

$$
\frac{\partial}{\partial t} C(t, t') = -\mu(t)\, C(t, t') + Q'(m(t))m(t') + \int_0^t R(t, t'') Q''(C(t, t'')) C(t', t'') dt''
$$

$$
+ \int_0^{t'} R(t', t'') Q'(C(t, t'')) dt'' , \tag{8}
$$

$$
\frac{\partial}{\partial t} R(t, t') = -\mu(t)\, R(t, t') + \int_{t'}^t R(t, t'') Q''(C(t, t'')) R(t'', t') dt'' , \tag{9}
$$

$$
\frac{d}{dt} m(t) = -\mu(t)\, m(t) + Q'(m(t)) + \int_0^t R(t, t'') m(t'') Q''(C(t, t'')) dt'' , \tag{10}
$$

$$
\mu(t) = Q'(m(t))m(t) + \int_0^t R(t, t'') \left[ Q'(C(t, t'')) + Q''(C(t, t'')) C(t, t'') \right] dt'' , \tag{11}
$$

with initial conditions $C(t, t) = 1\ \forall t$ and $R(t, t') = 0$ for all $t < t'$ and $\lim_{t' \to t^-} R(t, t') = 1\ \forall t$. The additional function $\mu(t)$, and its associated equation, are due to the spherical constraint; $\mu(t)$ plays the role of a Lagrange multiplier and guarantees that the solution of the previous equations is such that $C(t, t) = 1$. The derivation of these equations can be found in [23] and in the SM Sec. B. It is obtained using heuristic theoretical physics approach and can be very plausibly made fully rigorous generalising the work of [28, 36].

This set of equations can be solved numerically as described in [23]. The numerical estimation of the algorithmic threshold of gradient-flow, reproduced in Fig. 2, was obtained in [22]. We have also directly simulated the gradient flow Eq. (4) and compare the result to the one obtained from solving Eqs. (8-11). As shown in the SM Sec. C, for $N = 65535$, we find a very good agreement even for this large yet finite size.

**Surfing on saddles:** Armed with the dynamical equations, we now confirm the prediction of the threshold (5) based on the Kac-Rice-type of landscape analysis. In the SM we check that the minima trapping the dynamics are indeed the marginally stable ones ($t = 2$), see Figs. B.1 and B.2 in the SM, and we show the energy can be expressed in terms of $C$, $R$ and $m$. In the right panel of Fig. 4 we then plot the energy as a function of time obtained from the numerical solution of Eqs. (8-11) for $1/\Delta_2 = 1.5, 1.9, 2.3, 2.7$ and $\Delta_p = 1$ (same points and colour code of Figs. 2 and 3). For the two smaller values of $1/\Delta_2$ the energy converges to a plateau value at $\epsilon_{\text{th}}$ (dotted line), whereas for $1/\Delta_2 = 2.3, 2.7$ the energy plateaus close to $\epsilon_{\text{th}}$ but then eventually drifts away and reaches a lower value, corresponding to the global minimum correlated with the signal. This behaviour can be understood in terms of the spectral properties of the Hessian (6) of the minima trapping the dynamics.

In the left panel of Fig. 4 we plot the corresponding density of eigenvalues of $\mathcal{H}$ for the same values of $1/\Delta_2$ and $\Delta_p$ used in the right panel. This is an illustration of the dynamical phenomenon explained in the previous section: when the signal-to-noise ratio is large enough threshold minima become unstable because a negative eigenvalue, associated to a downward direction toward the signal, emerges. In this case $\boldsymbol{\sigma}(t)$ first seems to converge to the threshold minima and then, at long times, drifts away along the unstable direction. The larger is the signal-to-noise ratio the more unstable is the downward direction and, hence, the shortest is the intermediate trapping time.

### 3.2 Gradient-flow Threshold from Dynamical Theory

We now show that the very same prediction (5) for the algorithmic threshold of gradient-flow can be directly obtained analysing the dynamical equations (8-11), without directly using results from the Kac-Rice analysis, thus establishing a firm and novel connection between the behaviour of the gradient-flow algorithm and Kac-Rice landscape approaches.

For small signal-to-noise ratios, when $m$ remains zero at all times, the dynamical equations (8-11) are identical to the well-known one in spin glasses theory, for reviews see [37, 38]. These equations have been studied extensively for decades in statistical physics and a range of results about their behaviour has been established. Here we describe the results which are important for our analysis and devote the SM Sec. B.2 to a more extended presentation. It was shown analytically in [34] that the behaviour of the dynamics at large times is captured by an asymptotic solution of Eqs. (8-11) that verifies several remarkable properties. The ones of interest to us are that for $t$ and $t'$ large:

(i) $C(t, t') = 1$ when $t - t'$ finite; $C(t, t')$ becomes less than one when $t - t'$ diverges with $t$ and $t'$.

(ii) $R(t, t') = R_{\mathrm{TTI}}(t - t') + R_{\mathrm{ag}}(t, t')$, where TTI stands for time-translational-invariance, ag stand for aging. Here $R_{\mathrm{TTI}}(t - t')$ goes to zero on a finite time-scale, whereas $R_{\mathrm{ag}}(t, t')$ varies on timescales diverging with $t$ and $t'$. Moreover, $R_{\mathrm{ag}}(t, t')$ verifies the so called "weak-long term memory" property: for any finite $t_0$, $\int_{t-t_0}^{t} R_{\mathrm{ag}}(t, t'')dt''$ is arbitrarily small. We refer to this function form for $R(t, t')$ as the *aging ansatz*, adopting the physics terminology.

These properties are confirmed to hold by our numerical solution, see for instance Fig. B.3 in the SM. The interpretation of these dynamical properties is that at long times $\boldsymbol{\sigma}(t)$ decreases in the energy landscape and approaches the marginally stable minima. Concomitantly, dynamics slows down and takes place along the almost flat directions associated to the vanishing eigenvalues of the Hessian.

We remind that in the previous paragraphs we assume null correlation with the signal, $m = 0$. In order to find the algorithmic threshold beyond which the gradient-flow develops a positive correlation, we study the instability of the aging solution as a function of the signal-to-noise ratio. Our strategy is to start with an arbitrarily small overlap, $m(0) = \delta$, and determine whether it grows at long times thus indicating an instability towards the signal. Since the initial condition for the overlap is uncorrelated with the signal, then, for sufficiently small $\delta$, $C$ and $R$ reach their asymptotic form before $m$ becomes of order one. We can thus plug the asymptotic aging ansatz for $R$ in the dynamical equation for $m$:

$$\frac{d}{dt}m(t) = -\mu(t)m(t) + Q'(m(t)) + \int_0^t R_{\mathrm{TTI}}(t - t'')Q''(1)m(t'')dt'' + $$
$$+ \int_0^t R_{\mathrm{ag}}(t, t'')Q''(C_{\mathrm{ag}}(t, t''))m(t'')dt'' \tag{12}$$

In the linear approximation the solution has the form $m(t) = \delta \exp(\Lambda t)$ and we assume $\Lambda$ arbitrarily small since we want to find the algorithmic threshold where $\Lambda = 0$. The term $Q'(m(t))$ becomes $Q''(0)m(t)$. Since $m(t)$ has an arbitrarily slow evolution, whereas $R_{\mathrm{TTI}}(t - t'')$ relaxes to zero on a finite timescale, the second term of the RHS of eq. (12) simplifies to:

$$\delta \exp(\Lambda t)Q''(1)\int_0^t R_{\mathrm{TTI}}(t - t'')\exp(-\Lambda(t - t''))dt'' \simeq m(t)Q''(1)\overline{R}$$

where $\overline{R} = \int_0^t R_{\mathrm{TTI}}(t - t'')dt''$ does not depend on $t$ (since $t$ can be taken arbitrarily large and $R_{\mathrm{TTI}}(t - t'')$ relaxes to zero on finite time-scales). The contribution of to the last term on (12) reads:

$$\delta \exp(\Lambda t)\int_0^t R_{\mathrm{ag}}(t, t'')Q''(C_{\mathrm{ag}}(t, t''))\overline{e}^{-\Lambda(t - t'')}dt'' = m(t)\int_0^t R_{\mathrm{ag}}(t, t'')Q''(C_{\mathrm{ag}}(t, t''))\overline{e}^{-\Lambda(t - t'')})dt''.$$

Using that $Q''(C_{\text{ag}}(t, t''))$ is bounded by $Q''(1)$ and that $\Lambda$ cuts-off the integral on a time $t_0 \sim 1/\Lambda$ that does not diverge with $t$, we can use the "weak-long term memory" property to conclude that the last term is arbitrarily small compared to $m(t)$ and hence can be neglected with respect to the previous ones. Collecting all the pieces together we find:

$$\frac{d}{dt}m(t) = \left[-\mu_\infty + Q''(0) + Q''(1)\overline{R}\right] m(t) + O(\delta^2). \tag{13}$$

This is solved by $m(t) = \delta \exp(\Lambda t)$ with $\Lambda = -\mu_\infty + Q''(0) + Q''(1)\overline{R}$, which therefore justifies a posteriori our assumption of exponential growth. The condition for the instability of the aging solution towards the signal solution is therefore given by

$$0 = -\mu_\infty + Q''(0) + Q''(1)\overline{R}. \tag{14}$$

From the analysis of the asymptotic aging solution presented in SM Sec. B.2 one finds that $\mu_\infty = 2\sqrt{Q''(1)}$ and $\overline{R} = 1/\sqrt{Q''(1)}$, therefore obtaining $Q''(0) = \sqrt{Q''(1)}$. This condition is the same one found from the study of the landscape, and thus leads to the transition line eq. (5).

**Acknowledgments**

We thank Pierfrancesco Urbani for proof checking of the draft and many related discussions. We would also like to thank the Kavli Institute for Theoretical Physics (KITP) for welcoming us during part of this research, with the support of the National Science Foundation under Grant No. NSF PHY-1748958 We acknowledge funding from the ERC under the European Union's Horizon 2020 Research and Innovation Programme Grant Agreement 714608-SMiLe; from the European Union's Horizon 2020 research and innovation programme under the Marie Skłodowska-Curie grant agreement CoSP No 823748; from the French National Research Agency (ANR) grant PAIL; and from the Simons Foundation (#454935, Giulio Biroli).

## Footnotes

† Institut de Physique Théorique, CNRS & CEA & Université Paris-Saclay, Saclay, France.

‡ Laboratoire de Physique de l'Ecole normale supérieure ENS, Université PSL, CNRS, Sorbonne Université, Université Paris-Diderot, Sorbonne Paris Cité Paris, France.

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
