[Supplementary Material]

# Supplementary Material of
# Who is Afraid of Big Bad Minima? Analysis of Gradient-Flow in a Spiked Matrix-Tensor Model

**Stefano Sarao Mannelli**[†], **Giulio Biroli**[‡], **Chiara Cammarota**[∗],
**Florent Krzakala**[‡], **and Lenka Zdeborová**[†]

## Contents

† Institut de Physique Théorique, CNRS & CEA & Université Paris-Saclay, Saclay, France.

‡ Laboratoire de Physique de l'Ecole normale supérieure ENS, Université PSL, CNRS, Sorbonne Université, Université Paris-Diderot, Sorbonne Paris Cité Paris, France.

∗ Department of Mathematics, King's College London, Strand London WC2R 2LS, UK.

# A Kac-Rice method

## A.1 Summary of the Kac-Rice complexity

Figure A.1: Curves of the complexity of critical points, dotted and dashed curves from Eq. (A.3), and of minima, full curve from Eq. (A.5), at overlap value $m = 0$ at fixed $\Delta_p = 1.0$ for different $\Delta_2$. The figure shows qualitatively the same features as Fig. 3, but displays the full positive part of the complexity for the four cases discussed in the main text, $1/\Delta_2 \in \{1.5,\ 1.9,\ 2.3,\ 2.7\}$. Zooms of the curves of the annealed complexity of critical points and minima when they cross zero at negative loss are in the panels labelled from (a) to (d) for increasing $1/\Delta_2$.

In this section we introduce the Kac-Rice formula and we show how to reduce it to an explicit expression for the spiked matrix-tensor model. The Kac-Rice formula evaluates the expected number of critical points of a rough function subject to a number of conditions. For an inference problem it is interesting to focus on the expected number of critical points constrained to have of given loss and a given overlap with the ground truth. For convenience reasons we consider the rescaled loss $\mathcal{L}(\boldsymbol{\sigma}) = N\ell(\boldsymbol{\sigma})$. The Kac-Rice formula then reads

$$\mathbb{E}_\eta[\mathcal{N}(\epsilon, m|\Theta)] = \int_{\mathbb{S}^{N-1}} \delta\left(\langle\boldsymbol{\sigma}, \boldsymbol{\sigma}^*\rangle - m\right) \mathbb{E}_\eta\left[\left|\det\mathcal{H}\right|\middle|\mathcal{L} = N\,\epsilon, \partial_i\mathcal{L} = 0\ \forall i, \lambda_{\min} > 0\right] \times \tag{A.1}$$
$$\times\ \phi_{\mathcal{L}, \partial_i\mathcal{L}}(\boldsymbol{\sigma}, \mathbf{0}, \epsilon)d\boldsymbol{\sigma}\,,$$

where $\eta$ represents the noise in the problem, $\Theta$ the parameters and $\phi$ the joint probability density of the loss and its gradient.

The quantity of interest is the density of the logarithm of the number of critical points $\log\mathcal{N}(\epsilon, m|\Theta)/N$. It should be noted that, since the random variable representing the number of critical point fluctuates at the exponential scale, a correct estimation of the expected value of this quantity is not $\log\mathbb{E}_\eta[\mathcal{N}(\epsilon, m|\Theta)]$, as it would be immediately obtained by using the result of the Kac-Rice formula [1], but $\mathbb{E}_\eta[\log\mathcal{N}(\epsilon, m|\Theta)]$. These two quantities are called respectively *annealed* and *quenched complexities*. Using Jensen inequality one observes that the annealed complexity is just an upper bound of the quenched one. However, for mathematical convenience most of the studies have been focused on the former. Eventually the second moment of the number of critical points has been evaluated [2], by an extension of the Kac-Rice formula to higher moments [3], just to prove that the two are equivalent in some models [2]. The quenched complexity has been evaluated in a related model in a non rigorous way by studying the $n$-th moment and applying replica trick, the so-called

replicated Kac-Rice [4]. Given a random variable $Y$ replica trick says

$$\mathbb{E}_\eta[\log Y] = \lim_{n \to 0^+} \frac{\mathbb{E}_\eta[Y^n] - 1}{n} \tag{A.2}$$

but instead of considering an arbitrary $n \in \mathbb{R}^+$, the study is done using $n \in \mathbb{N}$ and performing an analytic continuation of the result to $0^+$. The replica trick has already been used in a plethora of applications and, although not rigorous, it was found correct in all naturally motivated cases that have been later approached by other techniques. An important mathematical literature has developed in order to understand the method.

In the next section we sketch the derivation of the quenched Kac-Rice and we provide all the information to determine the annealed one. Since the threshold is determined considering the configuration with arbitrarily small overlap $m \ll 1$, we focus on that case. Remarkably we found that as $m \to 0$ the quenched complexity is equal to the annealed one. We show that the corresponding Hessian is Eq. (6) in the main text, *i.e.* it is proportional to a GOE translated by $t$ and perturbed by a rank $n$ perturbation of strength $\theta$ that in the annealed case is of rank 1. Thus we find that the complexity for the stationary points is [1]

$$\Sigma_a^{\text{sta}}(m, \epsilon | \Delta_p, \Delta_2) = \max_{\substack{\epsilon_p, \epsilon_2 \\ \text{s.t. } \epsilon_p + \epsilon_2 = \epsilon}} \frac{1}{2} \log \frac{Q''(1)}{Q'(1)} + \frac{1}{2} \log(1 - m^2) - \frac{1}{2} \frac{(Q''(m))^2}{Q'(1)} (1 - m^2) + \tag{A.3}$$
$$- \frac{p\Delta_p}{2} \left( \epsilon_p + \frac{m^p}{p\Delta_p} \right)^2 - \Delta_2 \left( \epsilon_2 + \frac{m^2}{2\Delta_2} \right)^2 + \Phi(t),$$

with

$$\Phi(t) = \begin{cases} \frac{t^2}{4} & \text{if } |t| \leq 2 \\ \frac{t^2}{4} + \log \left( \sqrt{\frac{t^2}{4} - 1} + \frac{|t|}{2} \right) - \frac{|t|}{4} \sqrt{t^2 - 4} & \text{otherwise} \end{cases} \tag{A.4}$$

and $t = (p\epsilon_p + 2\epsilon_2)/\sqrt{Q''(1)}$ as already introduced in the main text. Finally studying the eigenvalue of the Hessian to constrain them in the positive semi-axis, we find the complexity of minima [1]

$$\Sigma_a(m, \epsilon | \Delta_p, \Delta_2) = \Sigma_a^{\text{sta}}(m, \epsilon_p, \epsilon_2 | \Delta_p, \Delta_2) - L(\theta, t). \tag{A.5}$$

with

$$L(\theta, t) = \begin{cases} \frac{1}{4} \int_{\theta + \frac{1}{\theta}}^t \sqrt{y^2 - 4} \, dy - \frac{\theta}{2} \left( t - \left( \theta + \frac{1}{\theta} \right) \right) + \frac{t^2 - \left( \theta + \frac{1}{\theta} \right)^2}{8} & \theta > 1, \ 2 \leq t < \frac{\theta^2 + 1}{\theta} \\ \infty & t < 2 \\ 0 & \text{otherwise.} \end{cases} \tag{A.6}$$

and $\theta = Q''(m)(1 - m^2)/\sqrt{Q''(1)}$. In Fig. A.1, we show the two complexities of the stationary points and of the minima in the parameter space discussed in the main text with discontinuous lines Eq. (A.3) and full lines Eq. (A.5), respectively. A positive complexity means an exponential number of critical points (minima). The region where exponentially many minima appear is highlighted in the small figures, showing the coexistence of exponentially many minima and saddles.

## A.2 Derivation of the quenched complexity

We proceed with the computation of the quenched Kac-Rice complexity for the spiked matrix-tensor model, using replicated Kac-Rice prescription for the spiked pure-tensor model [4]. This implies, following replica trick Eq. (A.2), the evaluation of the $n$-th moment of number of minima using Kac-Rice formula which is given by [3]

$$\mathbb{E}_\eta[\mathcal{N}(\epsilon, m | \Theta)^n] = \int_{\mathbb{S}^{N-1}} \cdots \int_{\mathbb{S}^{N-1}} \mathbb{E}_\eta \left[ \left( \prod_{a=1}^n |\det \mathcal{H}[\boldsymbol{\sigma}^a]| \right) \middle| \forall b, c \, \mathcal{L}[\boldsymbol{\sigma}^b] = N\epsilon, \partial_i \mathcal{L}[\boldsymbol{\sigma}^c] = 0 \, \forall i, \lambda_{\min} > 0 \right]$$
$$\times \phi_{\mathcal{L}, \partial_i \mathcal{L}} \left( \{\boldsymbol{\sigma}^a\}, \mathbf{0}, \epsilon \right) \prod_{a=1}^n \delta \left( \langle \boldsymbol{\sigma}^a, \boldsymbol{\sigma}^* \rangle - m \right) d\boldsymbol{\sigma}^a . \tag{A.7}$$

Where $\phi$ is the joint probability density of the loss and its gradients evaluated on the $n$ replicated configurations. We hereby sketch the main computation steps and present the results that are the most relevant for the theory presented in this paper, i.e. for $m = 0$. The details of the computation and further results for $m > 0$ will be presented in a dedicated work elsewhere.

It is convenient to consider free variables on $\mathbb{R}^N$ and constrain them using a Lagrange multiplier $\gamma$. Thus $\mathcal{L}(\boldsymbol{\sigma}^a) \mapsto \mathcal{L}(\boldsymbol{\sigma}^a) - \frac{\gamma}{2}\left(\sum_i (\sigma_i^a)^2 - 1\right)$. Using the fact that the gradient must be zero on the sphere, i.e. that $\nabla_i \mathcal{L}(\boldsymbol{\sigma}^a) - \gamma \sigma_i^a = 0 \;\; \forall i$, we obtain a simple expression for the multiplier: $\gamma = \langle \nabla \mathcal{L}(\boldsymbol{\sigma}^a), \boldsymbol{\sigma}^a \rangle$. Moreover by separating the two terms in the loss that represent the contribution of the two channels, $\mathcal{L} = \mathcal{L}_p + \mathcal{L}_2$, we define $\mathcal{L}_p = N\epsilon_p$ and $\mathcal{L}_2 = N\epsilon_2$ so to obtain for the multiplier the even simpler equation $\gamma = p\epsilon_p + 2\epsilon_2$. To take advantage of this simple formula, in the following we work with the contributions of the two channels to the loss function separately and we impose the constraint on their sum, $N\epsilon = N(\epsilon_p + \epsilon_2)$, only at the end.

The use of Cartesian coordinates allows us to evaluate easily the moments and covariances by means of standard derivatives.

$$\mathbb{E}_\eta\left[\mathcal{L}[\boldsymbol{\sigma}^a]\right] = -NQ(\langle \boldsymbol{\sigma}^a, \boldsymbol{\sigma}^* \rangle), \tag{A.8}$$

$$\mathrm{Cov}\left[\mathcal{L}[\boldsymbol{\sigma}^a], \mathcal{L}[\boldsymbol{\sigma}^b]\right] = N\,Q\left(\langle \boldsymbol{\sigma}^a, \boldsymbol{\sigma}^b \rangle\right). \tag{A.9}$$

Taking derivatives of these equation gives all the covariances of loss, gradient and Hessian. For instance, we can easily see that the covariance of the Hessian is given by:

$$\frac{\partial^4}{\partial \sigma_i^a \partial \sigma_j^a \partial \sigma_k^b \partial \sigma_l^b}\frac{\mathrm{Cov}\left[\mathcal{L}[\boldsymbol{\sigma}^a], \mathcal{L}[\boldsymbol{\sigma}^b]\right]}{N} = Q''''\left(\langle \boldsymbol{\sigma}^a, \boldsymbol{\sigma}^b \rangle\right)\langle \boldsymbol{\sigma}^b, \boldsymbol{e}_i^a \rangle\langle \boldsymbol{\sigma}^b, \boldsymbol{e}_j^a \rangle\langle \boldsymbol{\sigma}^a, \boldsymbol{e}_k^b \rangle\langle \boldsymbol{\sigma}^a, \boldsymbol{e}_l^b \rangle +$$

$$+ Q'''\left(\langle \boldsymbol{\sigma}^a, \boldsymbol{\sigma}^b \rangle\right)\left(\langle \boldsymbol{e}_i^a, \boldsymbol{e}_k^b \rangle\langle \boldsymbol{\sigma}^b, \boldsymbol{e}_j^a \rangle\langle \boldsymbol{\sigma}^a, \boldsymbol{e}_l^b \rangle + \langle \boldsymbol{e}_j^a, \boldsymbol{e}_k^b \rangle\langle \boldsymbol{\sigma}^b, \boldsymbol{e}_i^a \rangle\langle \boldsymbol{\sigma}^a, \boldsymbol{e}_l^b \rangle + \langle \boldsymbol{e}_i^a, \boldsymbol{e}_l^b \rangle\langle \boldsymbol{\sigma}^b, \boldsymbol{e}_j^a \rangle\langle \boldsymbol{\sigma}^a, \boldsymbol{e}_k^b \rangle +$$

$$+ \langle \boldsymbol{e}_j^a, \boldsymbol{e}_l^b \rangle\langle \boldsymbol{\sigma}^b, \boldsymbol{e}_i^a \rangle\langle \boldsymbol{\sigma}^a, \boldsymbol{e}_k^b \rangle\right) + Q''\left(\langle \boldsymbol{\sigma}^a, \boldsymbol{\sigma}^b \rangle\right)\left(\langle \boldsymbol{e}_i^a, \boldsymbol{e}_k^b \rangle\langle \boldsymbol{e}_j^a, \boldsymbol{e}_l^b \rangle + \langle \boldsymbol{e}_i^a, \boldsymbol{e}_l^b \rangle\langle \boldsymbol{e}_j^a, \boldsymbol{e}_k^b \rangle\right),$$

$$\tag{A.10}$$

where $\{\boldsymbol{e}_i^a\}_i$ and $\{\boldsymbol{e}_k^b\}_k$ are the reference frames associated to replica $a$ and $b$ respectively.

**Remark 1 (annealed Hessian)** *In particular notice that if $n = 1$ there is only one replica and using an orthogonal basis where the $N$-th direction is aligned with the replica and projecting on the sphere by discarding the last coordinates we obtain a simple expression:*

$$\frac{1}{N}Cov\left[\mathcal{H}_{ij}[\boldsymbol{\sigma}], \mathcal{H}_{kl}[\boldsymbol{\sigma}]\right] = Q''(1)\left(\delta_{ik}\delta_{jl} + \delta_{il}\delta_{jk}\right) \tag{A.11}$$

*with the delta representing Kronecker's deltas. This is the expression of a GOE. We can as well compute the mean deriving twice in the $i$-th and $j$-th coordinate. Following [5] we make another convenient choice for the basis imposing that the signal lies in the space spanned by the $\boldsymbol{e}_1$ and $\boldsymbol{e}_N = \boldsymbol{\sigma}$. This gives,*

$$\frac{1}{N}\mathbb{E}\left[\mathcal{H}_{ij}[\boldsymbol{\sigma}]\right] = Q''(\langle \boldsymbol{\sigma}^a, \boldsymbol{\sigma}^* \rangle)\langle \boldsymbol{\sigma}^*, \boldsymbol{e}_i \rangle\langle \boldsymbol{\sigma}^*, \boldsymbol{e}_j \rangle = Q''(\langle \boldsymbol{\sigma}^a, \boldsymbol{\sigma}^* \rangle)\langle \boldsymbol{\sigma}^*, \boldsymbol{e}_i \rangle\langle \boldsymbol{\sigma}^*, \boldsymbol{e}_j \rangle \delta_{i1}\delta_{j1} \tag{A.12}$$

*that, when $m = 0$, equals*

$$\frac{1}{N}\mathbb{E}\left[\mathcal{H}[\boldsymbol{\sigma}]\right] = Q''(0)\,\boldsymbol{e}_1\boldsymbol{e}_1^T. \tag{A.13}$$

*Wrapping together Eq. (A.11), Eq. (A.13) and the expression for the Langrange multiplier that acts as a translation, we obtain the Hessian presented in the main text Eq. (6). Observe, however, that the Hessian in which we are interested in is not the simple Hessian of the loss but rather the Hessian of the loss conditioned to a given loss and a given gradient. Using Eq. (A.9) to compute the covariance of Hessian and loss, and of Hessian and gradient under this basis, we can observe that these random variables are unconditioned. Thus the conditioning does not affect the distribution of the Hessian of the loss and therefore Eq. (6) is recovered.*

Eq. (A.8) and Eq. (A.9) are basic ingredients required to continue with the analysis. In the next two sections we first compute the joint density of the loss and its gradient, and second compute the expected value of the determinant of the Hessians. In the final section we put together the results obtaining the complexities already presented in the summary A.1.

### A.2.1 Joint probability density.

In order to evaluate the joint probability density $\phi$ we focus on the covariance matrix of the loss and its gradient, that using Eq. (A.9) is given by:

$$\frac{1}{N} \left[ \boldsymbol{C}_{\mathcal{L}, \nabla \mathcal{L}} \right]^{a,b} = \begin{bmatrix} Q'' \left( \langle \boldsymbol{\sigma}^a, \boldsymbol{\sigma}^b \rangle \right) \boldsymbol{\sigma}^a \otimes \boldsymbol{\sigma}^b + Q' \left( \langle \boldsymbol{\sigma}^a, \boldsymbol{\sigma}^b \rangle \right) \mathbb{I}_N & Q' \left( \langle \boldsymbol{\sigma}^a, \boldsymbol{\sigma}^b \rangle \right) \boldsymbol{\sigma}^{b,T} \\ Q' \left( \langle \boldsymbol{\sigma}^a, \boldsymbol{\sigma}^b \rangle \right) \boldsymbol{\sigma}^a & Q \left( \langle \boldsymbol{\sigma}^a, \boldsymbol{\sigma}^b \rangle \right) \end{bmatrix} . \tag{A.14}$$

The joint density corresponds to the probability of observing a zero gradient on the sphere and a given loss, $(\gamma \boldsymbol{\sigma}^T, \epsilon)^T$, in the multivariate Gaussian variable $(\nabla \mathcal{L}^T, \mathcal{L})^T$. Thus taking into account the first moments of loss and gradient, obtained from Eq. (A.8), we define the auxiliary vector $[\boldsymbol{\mu}(\epsilon_p, \epsilon_2)]^a = \left( (p\epsilon_p + 2\epsilon_2)\boldsymbol{\sigma}^{a,T} + Q'(\langle \boldsymbol{\sigma}^a, \boldsymbol{\sigma}^* \rangle)\boldsymbol{\sigma}^{*,T}, \ \epsilon + Q(\langle \boldsymbol{\sigma}^a, \boldsymbol{\sigma}^* \rangle) \right)^T$. The probability density is given by

$$\phi_{\mathcal{L}, \partial_i \mathcal{L}} \left( \{ \boldsymbol{\sigma}^a \}, \mathbf{0}, \epsilon \right) \propto \iint \delta(\epsilon - \epsilon_p - \epsilon_2) \exp \left[ -\frac{1}{2} \sum_{a,b} [\boldsymbol{\mu}(\epsilon_p, \epsilon_2)]^{a,T} \left[ \boldsymbol{C}_{\mathcal{L}, \nabla \mathcal{L}}^{-1} \right]^{a,b} [\boldsymbol{\mu}(\epsilon_p, \epsilon_2)]^b \right] d\epsilon_p d\epsilon_2. \tag{A.15}$$

This expression can be evaluated by observing that there is a set of $(N+1)n$-dimensional vectors that forms a closed group under the action of the covariance matrix Eq. (A.14). This set is composed by the following four vectors

$$\boldsymbol{\xi}_1^T = \left( \boldsymbol{\sigma}^{1,T}, 0, \boldsymbol{\sigma}^{2,T}, 0, \ldots, \boldsymbol{\sigma}^{n,T}, 0 \right) , \tag{A.16}$$

$$\boldsymbol{\xi}_2^T = \left( \sum_{e \neq 1} \boldsymbol{\sigma}^{e,T}, 0, \sum_{e \neq 2} \boldsymbol{\sigma}^{e,T}, 0, \ldots, \sum_{e \neq n} \boldsymbol{\sigma}^{e,T}, 0 \right) , \tag{A.17}$$

$$\boldsymbol{\xi}_3^T = \left( \mathbf{0}^T, 1, \mathbf{0}^T, 1, \ldots, \mathbf{0}^T, 1 \right) , \tag{A.18}$$

$$\boldsymbol{\xi}_4^T = \left( \boldsymbol{\sigma}^{*,T}, 0, \boldsymbol{\sigma}^{*,T}, 0, \ldots, \boldsymbol{\sigma}^{*,T}, 0 \right) , \tag{A.19}$$

where $\mathbf{0}$ is an $N$ dimensional null vector. Indeed the auxiliary vector can be rewritten in terms of the elements of this set of newly defined vectors as follows

$$[\boldsymbol{\mu}(\epsilon_p, \epsilon_2)]^a = (p\epsilon_p + 2\epsilon_2)[\boldsymbol{\xi}_1]^a + (\epsilon + Q(\langle \boldsymbol{\sigma}^a, \boldsymbol{\sigma}^* \rangle)) [\boldsymbol{\xi}_3]^a + Q'(\langle \boldsymbol{\sigma}^a, \boldsymbol{\sigma}^* \rangle) [\xi_4]^a . \tag{A.20}$$

At this point we exploit the fact that the set of these vectors forms a closed group under the action of the covariance matrix. In fact we can invert its action on the set $\{ \boldsymbol{\xi}_k \}_{k=1}^4$ only, without the need to evaluate the inverse of the full covariance matrix. Using this trick, the integrand in Eq. (A.15) can be evaluated. The result for the integrand in Eq. (A.15) contains the dependence on the configurations of replicas only in terms of the overlaps $q_{a,b} = \langle \boldsymbol{\sigma}^a, \boldsymbol{\sigma}^b \rangle$ with each other, and of the overlap of each of them with the ground truth, *i.e.* the magnetisation $m_a = \langle \boldsymbol{\sigma}^a, \boldsymbol{\sigma}^* \rangle$. In this formulation, hence, the problem of evaluating a free integral over $n$ vectors on the sphere has been translated into the task of evaluating an integral over the possible choices of the $n \times n$ matrix of the overlaps provided that we consider the multiplying factor that comes from the volume $V(\{q_{a,b}\}, \{m_a\})$ of configurations that are compatible with that choice and the condition on the magnetisations.

The next step is to make an ansatz on the form of the matrix of these overlaps which must be consistent with the condition on the vector of magnetisations required in the Kac-Rice formula. The simplest ansatz is called *replica symmetric* ansatz and assumes that the overlaps of different replicas are independent of the indices $a$ and $b$, i.e.

$$\langle \boldsymbol{\sigma}^a, \boldsymbol{\sigma}^b \rangle = \delta_{ab} + (1 - \delta_{ab}) \, q . \tag{A.21}$$

Note that the replica symmetric ansatz is compatible with the condition $\langle \boldsymbol{\sigma}^a, \boldsymbol{\sigma}^* \rangle = m \ \forall a$ imposed in the Kac-Rice formula. Within this ansatz the probability density can be evaluated as a function of $q$ and $m$ for arbitrary $n$ and the analytic continuation at $n \to 0^+$ can be finally taken to evaluate the quenched complexity. The expression for generic $n$ is too long and convoluted to be reported here. However in the limit $n \to 1$ it corresponds to the expression of the probability density of losses and

gradients evaluated in the annealed computation which has the following nice expression

$$\phi_{\mathcal{L}, \partial_i \mathcal{L}}\left(\{\boldsymbol{\sigma}^a\}, \mathbf{0}, \epsilon\right) \propto \int \int \delta(\epsilon - \epsilon_p - \epsilon_2) \exp\left[-\frac{N}{2} \frac{(Q''(m))^2}{Q'(1)}(1 - m^2)\right] \times$$

$$\times \exp\left[-\frac{Np}{2}\Delta_p\left(\epsilon_p + \frac{m^p}{p\Delta_p}\right)^2 - N\Delta_2\left(\epsilon_2 + \frac{m^2}{2\Delta_2}\right)^2\right] d\epsilon_p d\epsilon_2$$

$$\simeq \max_{\substack{\epsilon_p, \epsilon_2 \\ \text{s.t. } \epsilon_p + \epsilon_2 = \epsilon}} \exp\left[-\frac{N}{2}\frac{(Q''(m))^2}{Q'(1)}(1 - m^2) - \frac{Np}{2}\Delta_p\left(\epsilon_p + \frac{m^p}{p\Delta_p}\right)^2 - N\Delta_2\left(\epsilon_2 + \frac{m^2}{2\Delta_2}\right)^2\right].$$
(A.22)

We must also consider the normalisation of the density that is given by

$$\exp\left[-\frac{Nn}{2}\log(2\pi Q'(1))\right].$$
(A.23)

Finally we come back to the volume term $V(\{q_{a,b}\}, \{m_a\})$. Constraining the configurations to a given overlap $q$ with each other and $m$ with the ground truth produces a volume term that can be easily evaluated as

$$V(q, m) \simeq \exp\left[\frac{Nn}{2}\left(\log\frac{2\pi e(1 - q)}{N} - \frac{m^2 - q}{1 - q}\right)\right],$$
(A.24)

and for one single replica (which is useful in the computation of the annealed complexity) is simply

$$V(m) \simeq \exp\left[\frac{N}{2}\left(\log\frac{2\pi e}{N} + \log(1 - m^2)\right)\right].$$
(A.25)

Under the replica symmetric assumption we make a Laplace approximation that allows to evaluate the quenched complexity as an extremisation of the entire expression that depends on the overlap variable $q$ through the volume term $V(q, m)$ and the probability density $\phi$. An interesting remark concerns the limit $q \to 0$ in quenched joint probability density. Indeed in that case the two joint probability coincide module a factor $n$. We checked numerically that as $m \to 0$ the optimal $q$ goes to 0, which implies that the equations of the annealed and quenched complexities do correspond on the equator $m = 0$.

### A.2.2 Expected value of the Hessian.

As discussed introducing Eq. (A.9), the Hessian is a matrix-valued random variable with multivariate Gaussian distribution. In evaluating the Kac-Rice formula we must consider the distribution of the Hessian conditioned to the loss and its gradient, this can be imposed using the formula for conditioning of Gaussian random variables. Given $\boldsymbol{X}, \boldsymbol{Y}$ Gaussian random variables with covariance $\boldsymbol{C}$ and mean $\boldsymbol{\mu}$ the distribution of $\boldsymbol{X}$ conditioned to $\boldsymbol{Y} = \boldsymbol{y}^*$ is still Gaussian with covariance and mean

$$\boldsymbol{C}_{\boldsymbol{X}|\boldsymbol{Y}=\boldsymbol{y}^*} = \boldsymbol{C}_{\boldsymbol{X}} - \boldsymbol{C}_{\boldsymbol{X}\boldsymbol{Y}}\boldsymbol{C}_{\boldsymbol{Y}}^{-1}\boldsymbol{C}_{\boldsymbol{X}\boldsymbol{Y}};$$

$$\boldsymbol{\mu}_{\boldsymbol{X}|\boldsymbol{Y}=\boldsymbol{y}^*} = \boldsymbol{\mu}_{\boldsymbol{X}} + \boldsymbol{C}_{\boldsymbol{X}\boldsymbol{Y}}\boldsymbol{C}_{\boldsymbol{Y}}^{-1}(\boldsymbol{y}^* - \boldsymbol{\mu}_{\boldsymbol{Y}}).$$

In the annealed case, by using Eq. (A.9) and the expression for the Langrange multiplier $\gamma$ we get that the Hessian corresponds to a shifted GOE subject to a rank one perturbation, as already discussed in the main text (see Eq. (6)). In the replicated Kac-Rice formula a more complicated expression appears that depends on the product of the determinants of Hessians associated to different replicas. However, using a proper reference frame, it was already noticed [2, 4] that each Hessian corresponds also to a GOE since it is dominated by a $(N - n) \times (N - n)$ GOE block as $n \ll N$. Moreover it has also been shown [2, 4] that the expectation value of the product the Hessian determinants is equivalent to the product of the expectation values of each determinant. Thus we can still use standard results on the spectrum of GOE random matrices to evaluate the term in the Kac-Rice that depends on the Hessian. The distribution of the spectrum of the eigenvalue is given by

$$\rho(\lambda)d\lambda = \frac{\sqrt{4Q''(1) - (\lambda + \gamma)^2}}{2\pi Q''(1)}d\lambda,$$
(A.26)

Figure A.2: Shift of the Hessian, from Eq. (6), as function the loss density for different values of $m$. The qualitative behaviour shown in the figure does not change varying the parameter of the systems, i.e. it is always a decreasing function. The figure shows results obtained using $p = 3$, $\Delta_p = 1.0$ and $1/\Delta_2 = 2.3$.

thus the determinant is given by

$$\int \rho(\lambda) \log |\lambda| \, d\lambda = \frac{1}{2} \log[2Q''(1)] + \frac{1}{\pi} \int \sqrt{2 - \lambda^2} \log \left| \lambda - \gamma/\sqrt{Q''(1)} \right| d\lambda \,. \qquad \text{(A.27)}$$

where we recognise $t = \gamma/\sqrt{Q''(1)}$. After some algebra we find:

$$\frac{1}{2} \log[Q''(1)] - \frac{1}{2} + \Phi(t) \qquad \text{(A.28)}$$

with $\Phi(t)$ defined in Eq. (A.4).

### A.2.3  Complexities: Putting pieces together

By putting the above pieces together we obtain the annealed complexity of stationary points Eq. (A.3) where we can finally distinguish the origin of the various terms: the first term comes from the normalisation of the density and the determinant of the Hessian, the second comes from the volume prefactor, the third, fourth and fifth terms are originated by the probability density of loss and gradient and the last term comes from the product of Hessians.

In order to select only the minima in the study of the complexity we must impose that the smallest eigenvalue is positive. There are two possible scenarios: either the smallest eigenvalue is determined by the left edge of the bulk of the spectrum (the perturbation does not induces any BBP transition), or it is outside the bulk of the spectrum. In the first case the probability that the smallest eigenvalue is positive is suppressed by a factor $e^{-N^2}$ and the corresponding large deviation function is infinite. In the second case the large deviation function associated to the shift in the position of the smallest eigenvalue, that would allow to keep it positive, is finite and must be evaluated. The problem can be addressed with a replica computation [4] and focuses only on the typical value of the eigenvalue, missing the large deviation function. As we already discussed as $m \to 0$ we found numerically that the overlap $q$ that extremises the complexity is $q = 0$, which leads back to the annealed complexity as we have shown computing the density. Since the main focus in the paper is on the critical points at the equator we do not compute the isolated eigenvalue in a quenched approach but we rather use the large deviation function as it can be obtained in the annealed approximation [6] of which we report here the result. The condition on having a positive minimum eigenvalue suppresses the number of critical points by a factor $e^{-NL(\theta,t)}$, with $L(\theta,t)$ given in Eq. (A.6), that enters in Eq. (A.3) leading to Eq. (A.5).

**Remark 2 (threshold loss)** *Stefano TODO The derivation I was making is wrong.. this must be updated!!!*

# B  CHSCK Equations

In this section we present a derivation of CHSCK equations for the spiked matrix-tensor model using the generating functional formalism and later the asymptotic analysis under the hypothesis presented in the main text. The starting point is the loss, Eq. (3), expliciting the observations, Eqs. (1-2),

$$
\ell(\boldsymbol{\sigma}|\boldsymbol{T},\boldsymbol{Y}) = -\frac{\sqrt{(p-1)!}}{\Delta_p\sqrt{N}} \sum_{i_1<\cdots<i_p} \eta_{i_1\ldots i_p}\sigma_{i_1}\ldots\sigma_{i_p} - \frac{1}{p\Delta_p}\langle\boldsymbol{\sigma},\boldsymbol{\sigma}^*\rangle^p +
$$
$$
-\frac{1}{\Delta_2\sqrt{N}}\sum_{i<j}\eta_{ij}\sigma_i\sigma_j - \frac{1}{2\Delta_2}\langle\boldsymbol{\sigma},\boldsymbol{\sigma}^*\rangle^2
\tag{B.1}
$$

and the gradient flow Eq. (4) that for mathematical convenience we associate to an auxiliary function $\boldsymbol{f}(\boldsymbol{\sigma})$

$$
\dot{\sigma}_i(t) = -\mu(t)\sigma(t) - \frac{\partial\ell(\boldsymbol{\sigma}(t)|\boldsymbol{T},\boldsymbol{Y})}{\partial\sigma_i(t)} \doteq f_i(\boldsymbol{\sigma}(t)) .
\tag{B.2}
$$

The next section shows in detail the derivation that proceeds by introducing a probability distribution for the different evolutions, or trajectories, of the dynamics at a fixed realization of the noise. Then the distribution is averaged over the noise and this implies some technical steps before obtaining the final form. The resulting distribution is used to average correlation, response function and magnetisation over all the trajectories giving the CHSCK Eqs. (8-11). In the analysis an important role is played by the normalisation constant of the distribution of the trajectories, that is used in the final steps to derive with simplicity the equations.

After deriving the equations we show how to apply the hypothesis on the large time behaviour of $t$ and $t'$ to the CHSCK Eqs. In the last part this analysis provides the constants $\overline{R}$ and $\mu_\infty$ used to the derive the threshold in the main text, and some interesting additional information, such as the value of the loss at the threshold shown in the right panel of Fig. 4.

## B.1  Derivation of CHSCK Equations

The first step is to discretise the time in $M$ time steps of length $h$. We want the trajectories to be a solution at every time step of Eq. (4), which discretized looks as $\sigma_i^{a+1} - \sigma_i^a = f_i(\boldsymbol{\sigma}^a)h$ with $a$ the time index. Let's call $y^{a+1}$ a solution to this equation. We can define the probability density of observing a trajectory satisfying Eq. (4) at a fixed noise:

$$
p(\boldsymbol{\sigma}^1,\ldots,\boldsymbol{\sigma}^M) = \int \prod_{ai} \delta\left(\sigma_i^{a+1} - y_i^a(\boldsymbol{\sigma}^a)\right) \prod_{a=0}^{M-1} d\mu_{\mathbb{S}}^a .
\tag{B.3}
$$

where $\mu_{\mathbb{S}}$ is the measure over $\mathbb{S}^{N-1}$.

The normalisation constant is the integral of this probability and is called *generating functional* $\mathcal{Z}$ and since the previous object is already properly normalised it is equal to 1. Rewriting the $\delta$s as Fourier transforms and therefore including the auxiliary variables $\tilde{\boldsymbol{\sigma}}^a$,

$$
1 = \int \prod_{ai} \exp\left[N\tilde{\sigma}_i^a\left(\sigma_i^{a+1} - \sigma_i^a - f_i(\boldsymbol{\sigma}^a)h\right)\right] \prod_{a=0}^{M-1} d\mu_{\mathbb{S}}^a \frac{d\tilde{\boldsymbol{\sigma}}^a}{2\pi i}
\tag{B.4}
$$

where in order to have mathematically well-defined quantities in the large $N$ limit we have a factor in the exponential. Moving to the continuum, the generating functional appears as a path integral

$$
1 = \mathcal{Z} = \int \mathcal{D}[\boldsymbol{\sigma},\tilde{\boldsymbol{\sigma}}] \prod_i \exp\left[N\int \tilde{\sigma}_i(t)\left(\partial_t\sigma_i(t) - f_i(\boldsymbol{\sigma}(t))\right)dt\right] .
\tag{B.5}
$$

So far the object we derived is a distribution that tells whether a trajectory from arbitrary initial condition respects or not gradient-flow dynamics, however, our interest is in average trajectories with

respect to the realization of the disorder. Therefore the distribution has to be averaged and after some algebraic manipulation gives the average generating functional in Eq. (B.7),

$$
1 = \mathbb{E}_\eta\left[\mathcal{Z}\right] = \int \mathcal{D}\left[\boldsymbol{\sigma},\tilde{\boldsymbol{\sigma}}\right] \prod_i \exp\left[N\int \tilde{\sigma}_i(t)\left(\partial_t\sigma_i(t) + \mu(t)\sigma_i(t) - Q'(\langle\boldsymbol{\sigma}(t),\boldsymbol{\sigma}^*\rangle^{p-1})\sigma_i^*\right)dt\right] \times
$$

$$
\times \mathbb{E}_\eta\left\{\prod_i \exp\left[-\int\tilde{\sigma}_i(t)\left(-\frac{\sqrt{N(p-1)!}}{\Delta_p}\sum_{i_1<\cdots<i_p}\eta_{i\,i_1\ldots i_{p-1}}\sigma_{i_1}\ldots\sigma_{i_{p-1}}\right)dt\right]\right\} \times
$$

$$
\times \mathbb{E}_\eta\left\{\prod_i \exp\left[-\int\tilde{\sigma}_i(t)\left(-\frac{\sqrt{N}}{\Delta_2}\sum_j\eta_{ij}\sigma_j\right)dt\right]\right\}.
$$

(B.6)

In averaging over the $\eta$ we need to be careful in grouping all the permutations of $i$ with $i_1,\ldots,i_{p-1}$. For instance the exponent of the term in $p$ is gives by

$$
-\frac{\sqrt{N(p-1)!}}{\Delta_p}\sum_{i_1<\cdots<i_p}\int\eta_{i_1\ldots i_p}\left(\tilde{\sigma}_{i_1}(t)\sigma_{i_2}(t)\ldots\sigma_{i_p}(t) + \cdots + \sigma_{i_1}(t)\sigma_{i_2}(t)\ldots\tilde{\sigma}_{i_p}(t)\right)dt
$$

$$
= \frac{N(p-1)!}{2\Delta_p}\sum_{i_1<\cdots<i_p}\iint\left(\tilde{\sigma}_{i_1}(t)\sigma_{i_2}(t)\ldots\sigma_{i_p}(t) + \text{perm.}\right)\left(\tilde{\sigma}_{i_1}(t')\sigma_{i_2}(t')\ldots\sigma_{i_p}(t') + \text{perm.}\right)dtdt'
$$

$$
= \frac{N}{2\Delta_p}\iint\left(\langle\tilde{\boldsymbol{\sigma}}(t),\tilde{\boldsymbol{\sigma}}(t')\rangle\langle\boldsymbol{\sigma}(t),\boldsymbol{\sigma}(t)\rangle^{p-1} + (p-1)\langle\tilde{\boldsymbol{\sigma}}(t),\boldsymbol{\sigma}(t')\rangle\langle\boldsymbol{\sigma}(t),\tilde{\boldsymbol{\sigma}}(t')\rangle\langle\boldsymbol{\sigma}(t),\boldsymbol{\sigma}(t)\rangle^{p-2}\right)dtdt'.
$$

This gives an action $\mathcal{S}[\boldsymbol{\sigma},\tilde{\boldsymbol{\sigma}}]$ defined by

$$
1 = \overline{\mathcal{Z}} = \int\mathcal{D}[\boldsymbol{\sigma},\tilde{\boldsymbol{\sigma}}]e^{\mathcal{S}[\boldsymbol{\sigma},\tilde{\boldsymbol{\sigma}}]} =
$$

$$
= \int\mathcal{D}\left[\boldsymbol{\sigma},\tilde{\boldsymbol{\sigma}}\right]\prod_i\exp\left[N\int\tilde{\sigma}_i(t)\left(\partial_t\sigma_i(t) + \mu(t)\sigma_i(t) - Q'(\langle\boldsymbol{\sigma}(t),\boldsymbol{\sigma}^*\rangle^{p-1})\sigma_i^*\right)dt\right] \times
$$

$$
\times \exp\left[\frac{N}{2\Delta_p}\iint\langle\tilde{\boldsymbol{\sigma}}(t),\tilde{\boldsymbol{\sigma}}(t')\rangle\langle\boldsymbol{\sigma}(t),\boldsymbol{\sigma}(t)\rangle^{p-1}dtdt'\right] \times
$$

$$
\times \exp\left[\frac{N}{2\Delta_p}\iint(p-1)\langle\tilde{\boldsymbol{\sigma}}(t),\boldsymbol{\sigma}(t')\rangle\langle\boldsymbol{\sigma}(t),\tilde{\boldsymbol{\sigma}}(t')\rangle\langle\boldsymbol{\sigma}(t),\boldsymbol{\sigma}(t)\rangle^{p-2}dtdt'\right] \times
$$

$$
\times \exp\left[\frac{N}{2\Delta_2}\iint\left(\langle\tilde{\boldsymbol{\sigma}}(t),\tilde{\boldsymbol{\sigma}}(t')\rangle\langle\boldsymbol{\sigma}(t),\boldsymbol{\sigma}(t)\rangle + \langle\tilde{\boldsymbol{\sigma}}(t),\boldsymbol{\sigma}(t')\rangle\langle\boldsymbol{\sigma}(t),\tilde{\boldsymbol{\sigma}}(t')\rangle\right)dtdt'\right].
$$

(B.7)

A simple way to proceed once evaluated the action was proposed in [7] and consists in taking the expectation with respect to the path distribution and exploiting simple identities together with integration by part:

$$
0 = -\left\langle\frac{\delta\sigma_i(t')}{\delta\tilde{\sigma}_i(t)}\right\rangle_\mathcal{S} = \left\langle\sigma_i(t')\frac{\delta\mathcal{S}}{\delta\tilde{\sigma}_i(t)}\right\rangle_\mathcal{S} =
$$

$$
= N\Bigg\langle\partial_t\sigma_i(t)\sigma_i(t') + \mu(t)\sigma_i(t)\sigma_i(t') - Q'(\langle\boldsymbol{\sigma}(t),\boldsymbol{\sigma}^*\rangle^{p-1})\sigma_i^*\sigma_i(t') +
$$

$$
+ \frac{1}{\Delta_p}\int\left[\langle\boldsymbol{\sigma}(t),\boldsymbol{\sigma}(t)\rangle^{p-1}\tilde{\sigma}_i(t'') + (p-1)\langle\tilde{\boldsymbol{\sigma}}(t),\boldsymbol{\sigma}(t'')\rangle\langle\boldsymbol{\sigma}(t),\boldsymbol{\sigma}(t)\rangle^{p-2}\sigma_i(t'')\right]dt'' +
$$

$$
+ \frac{1}{\Delta_2}\int\left[\langle\boldsymbol{\sigma}(t),\boldsymbol{\sigma}(t)\rangle\tilde{\sigma}_i(t'') + \langle\tilde{\boldsymbol{\sigma}}(t),\boldsymbol{\sigma}(t'')\rangle\sigma_i(t'')\right]dt''\Bigg\rangle_\mathcal{S}.
$$

(B.8)

Figure B.1: $t = -(p\epsilon_p(t) + 2\epsilon_2(t))/\sqrt{Q''(1)}$ for $p = 3$, $\Delta_p = 1.0$ and $1/\Delta_2 = 1.9$ evaluated numerically from the CHSCK equations.

Finally, summing over the index $i$ and dividing by $N$ we recover Eq. (8). The remaining CHSCK Eqs. (9-10) follow analogously from:

$$\delta(t - t') = \sum_i \left\langle \frac{\delta\tilde{\sigma}_i(t')}{\delta\tilde{\sigma}_i(t)} \right\rangle_{\mathcal{S}} = -\sum_i \left\langle \tilde{\sigma}_i(t') \frac{\delta S}{\delta\tilde{\sigma}_i(t)} \right\rangle_{\mathcal{S}} ; \tag{B.9}$$

$$0 = -\left\langle \frac{\delta\sigma_i^*}{\delta\tilde{\sigma}_i(t)} \right\rangle_{\mathcal{S}} = \left\langle \sigma_i^* \frac{\delta S}{\delta\tilde{\sigma}_i(t)} \right\rangle_{\mathcal{S}} . \tag{B.10}$$

and Eq. (11) comes from imposing the spherical constrain, $C(t, t) = 1 \quad \forall t$, on Eq. (8).

In the following we are going to perform the analysis proposed by [8] in the present model. We need to consider Langevin dynamics instead of gradient-flow dynamics

$$\dot{\sigma}_i(t) = -\mu(t)\sigma(t) - \frac{\partial\ell(\boldsymbol{\sigma}(t)|\boldsymbol{T}, \boldsymbol{Y})}{\partial\sigma_i(t)} + \frac{1}{\sqrt{N}}\eta_i^{(L)}(t), \tag{B.11}$$

where the last term represents the Langevin noise, which is white Gaussian noise with moments: $\mathbb{E}_L[\eta_i^{(L)}(t)] = 0$ and $\mathbb{E}_L[\eta_i^{(L)}(t)\eta_j^{(L)}(t')] = 2T\delta_{ij}\delta(t - t')$ with $T$ that has the physical meaning of temperature. The CHSCK equations are slightly modified,

$$\frac{\partial}{\partial t}C(t, t') = TR(t', t) - \mu(t)\,C(t, t') + Q'(m(t))m(t') +$$

$$+ \int_0^t R(t, t'')Q''(C(t, t''))C(t', t'')dt'' + \int_0^{t'} R(t', t'')Q'(C(t, t''))dt'' , \tag{B.12}$$

$$\frac{\partial}{\partial t}R(t, t') = -\mu(t)\,R(t, t') + \int_{t'}^t R(t, t'')Q''(C(t, t''))R(t'', t')dt'' , \tag{B.13}$$

$$\frac{\partial}{\partial t}m(t) = -\mu(t)\,m(t) + Q'(m(t)) + \int_0^t R(t, t'')m(t'')Q''(C(t, t''))dt'' , \tag{B.14}$$

$$\mu(t) = T + Q'(m(t))m(t) + \int_0^t R(t, t'')\left[Q'(C(t, t'')) + Q''(C(t, t''))C(t, t'')\right]dt'' . \tag{B.15}$$

## B.2  CHSCK Equations Separation of Time-Scales

The theory of glassy dynamics [9] is quite involved. We have therefore decided to show first some numerical results that directly confirm assumptions made in the main text, and then show in full glory that these assumptions can be obtained analytically from the theory.

### B.2.1  Numerical tests

The first property that we wish to test is that for low signal-to-noise ratio GF is trapped in minima that are marginally stable. This can be checked computing from the CHSCK equations the evolution of

Figure B.2: Support of the density of eigenvalues of the Hessian along the dynamics for $p = 3$, $\Delta_p = 1.0$ and $1/\Delta_2 = 1.9$. This figure illustrates the GF tends to the marginally stable minima for low signal-to-noise ratio.

Figure B.3: The correlation function $C(t, t')$ with $p = 3$, $\Delta_p = 1.0$ and $1/\Delta_2 = 1.5$ evaluated numerically from the CHSCK equations. The correlation is plotted as difference of the two times showing the as $t - t' \ll t, t'$ it remains close to 1. This shows as well that in this regime the correlation function shows time translational invariance.

$t = -\left(p\epsilon_p(t) + 2\epsilon_2(t)\right)/\sqrt{Q''(1)}$, the terms $\epsilon_p(t)$ and $\epsilon_2(t)$ can be expressed in terms of $C, R, m$ as shown in Rmk. 6 of section B.2.6. In Fig. B.1 we show that $t = -\left(p\epsilon_p(t) + 2\epsilon_2(t)\right)/\sqrt{Q''(1)}$ ($\epsilon_p(t)$ starts from zero at initial time and then converges to two. Thus the minima to which GF tends to at long times and for small signal-to-noise ratio are indeed the *marginally stable ones* characterized by a spectrum of the Hessian whose left edge touches zero. Actually, transferring the results obtained in the context of spin-glasses to our case [10], we know that as long as $m$ remains zero, i.e for small signal-to-noise ratio, the spectrum of the Hessian along the dynamics is a Wigner semi-circle with support $[\sqrt{Q''(1)}(-2 + t), \sqrt{Q''(1)}(2 + t)]$. We show the evolution of the support as a function of time in Fig. B.2. This is another illustration of the fact that minima trapping GF are marginally stable.

The other point we wish to test is the assumption (i) made in the main text on $C(t, t')$, which we repeat here for convenience: $C(t, t') = 1$ when $t - t'$ finite; $C(t, t')$ becomes less than one when $t - t'$ diverges with $t$ and $t'$. We show in Fig. B.3 the correlation function $C(t, t')$ as a function of $t - t'$ (in log-scale) for several values of $t'$. This is a good illustration of the "aging ansatz" defined in the main text.

Let us stress that these numerical tests where already done in the past on similar spin-glass models. We show them here in order to make the paper self-contained and so that the reader does not have to go back to physics literature. In the same vein, in the next sections we show the full theoretical

analysis of the dynamical equations, which closely follows the theory of glassy dynamics developed in physics [9].

### B.2.2  General aging ansatz

We study the behaviours of the dynamics at large times in the two-time regimes introduced in the main text (now generalized at finite temperature).

1. $t, t' \gg 1$ with $\frac{t-t'}{t} \to 0$, see Fig. B.3. In this regime we have two important aspects: the two-times function depends only on the difference of the two times, $\tau = t - t'$, and we say that they respect time-translational invariance. Under this observation we redefine the two functions as $C(t,t') \to C_{\mathrm{TTI}}(\tau) \equiv C(t-t',0)$ and $R(t,t') \to R_{\mathrm{TTI}}(\tau) \equiv R(t-t',0)$. The second important aspect is the validity of the fluctuation-dissipation theorem (FDT) that links correlation and response function by the relation $R_{\mathrm{TTI}}(\tau) = -\frac{1}{T}\frac{dC_{\mathrm{TTI}}}{d\tau}(\tau)$ for $\tau$ positive.

2. $t, t' \gg 1$ with $\frac{t-t'}{t} = O(1)$. In this regime the relevant variable to consider is $\lambda = \frac{t'}{t}$. In reason of the "weak-long term memory" property it is useful to redefine rescale the response function and define $\mathcal{R}(\lambda) = tR(t,t')$. It is also convenient to consider the function $q\mathcal{C}(\lambda) = C(t,t')$ with $q = \lim_{\tau\to\infty} C_{\mathrm{TTI}}(\tau)$. Finally, in this regime a generalised version of the fluctuation-dissipation theorem holds $\mathcal{R}(\lambda) = \frac{1}{T}xq\frac{d\mathcal{C}(\lambda)}{d\lambda}$ where $x$ is called *violation parameter*.

Under the (generalized) FDT the equations for correlation and response function that we obtain in the two-time regime collapse into a single equation. In the first regime we analyse only the correlation because of this link, while in the second regime we consider the two equations separately since we need to determine the violation parameter $x$.

In the analysis that follows we use massively the hypothesis of the two regimes to split the integrals and analyse them separately. We start analysing the single time equation for the Lagrange multiplier $\mu(t)$, then we proceed with the two-times function concentrating first on the time-translational invariant part and then on the aging part.

### B.2.3  Langrange multiplier in the large time limit.

Starting from Eq. (B.15), we introduce the symbol $\clubsuit_p$ to isolate the two contribution of matrix and tensor to the integral. As the time tends to infinity $m$ and $\mu$ tend to their asymptotic value, respectively $m_\infty$ and $\mu_\infty$, Eq. (B.15) tends to

$$\mu_\infty = T + Q'(m_\infty)m_\infty + p\clubsuit_p + 2\clubsuit_{p=2}\,.$$

We can now use the idea of the separation in two-time regimes. Call $Q_p(x) = x^p/(p\Delta_p)$ the part related to $p$ in $Q(x)$,

$$\clubsuit_p = \int_0^t Q_p'(C(t,t''))R(t,t'')dt'' = \int_{\mathrm{FDT}} + \int_{\mathrm{aging}} =$$

$$= -\int_t^0 Q_p'(C(t,t-\tilde{t}))R(t,t-\tilde{t})d\tilde{t} + \int_0^1 \mathcal{R}(\lambda)Q_p'(q\mathcal{C}(\lambda))d\lambda =$$

$$= +\int_0^\infty Q_p'(C_{\mathrm{TTI}}(\tilde{t}))R_{\mathrm{TTI}}(\tilde{t})d\tilde{t} + \int_0^1 \mathcal{R}(\lambda)Q_p'(q\mathcal{C}(\lambda))d\lambda =$$

$$= -\int_0^\infty \frac{1}{T}\frac{d}{d\tilde{t}}Q_p(C_{\mathrm{TTI}}(\tilde{t}))d\tilde{t} + \int_0^1 \mathcal{R}(\lambda)Q_p'(q\mathcal{C}(\lambda))d\lambda =$$

$$= \frac{1-q^p}{T\Delta_p} + \int_0^1 \mathcal{R}(\lambda)Q_p'(q\mathcal{C}(\lambda))d\lambda$$

the resulting equation is

$$\mu_\infty = T + Q'(m_\infty)m_\infty + \frac{1}{T}\left[Q'(1) - qQ'(q)\right] +$$

$$+ p\int_0^1 \mathcal{R}(\lambda)Q_p'(q\mathcal{C}(\lambda))d\lambda + 2\int_0^1 \mathcal{R}(\lambda)Q_2'(q\mathcal{C}(\lambda))d\lambda\,. \tag{B.16}$$

### B.2.4 Regime 1: FDT.

We apply the same scheme of separating the times scale and applying FDT to the correlation function. All over the analysis we isolate terms in the equations using the symbols ♣ and ♠. Eq. (B.12) in the large time is

$$(\partial_\tau + \mu_\infty)C_{\text{TTI}}(\tau) = Q'(m_\infty)m_\infty + \spadesuit + \clubsuit, \tag{B.17}$$

with:

$$\begin{aligned}
\clubsuit &= \int_0^{t'} Q'(C(t,t''))R(t',t'')dt'' = \int_{\text{FDT}} + \int_{\text{aging}} = \\
&= -\int_{t'}^0 Q'(C(t,t'-\tilde{t}))R(t',t'-\tilde{t})d\tilde{t} + \int_0^1 Q'(qC(\lambda))\mathcal{R}(\lambda)d\lambda = \\
&= -\frac{1}{T}\int_0^\infty Q'(C_{\text{TTI}}(\tau+\tilde{t}))\frac{d}{d\tilde{t}}C_{\text{TTI}}(\tilde{t})d\tilde{t} + \int_0^1 Q'(qC(\lambda))\mathcal{R}(\lambda)d\lambda
\end{aligned}$$

and

$$\begin{aligned}
\spadesuit &= \int_0^t Q''(C(t,t''))R(t,t'')C(t',t'')dt'' = \int_{t'}^t + \int_0^{t'} = \int_{t'}^t + \int_{\text{FDT}} + \int_{\text{aging}} = \\
&= \frac{1}{T}\left[Q'(1)C_{\text{TTI}}(\tau) - Q'(q)q\right] - \frac{1}{T}\int_0^\tau Q'(C_{\text{TTI}}(\tau-\tilde{t}))\frac{d}{d\tilde{t}}C_{\text{TTI}}(\tilde{t})d\tilde{t} \\
&\quad + \frac{1}{T}\int_0^\infty Q'(C_{\text{TTI}}(\tau+\tilde{t}))\frac{d}{d\tilde{t}}C_{\text{TTI}}(\tilde{t})d\tilde{t} + \int_0^1 \mathcal{R}(\lambda)Q''(qC(\lambda))\,qC(\lambda)d\lambda\,.
\end{aligned}$$

Substituting these expressions and using Eq. (B.16) in Eq. (B.17) we have the first equation, which characterises the first regime

$$\partial_\tau C_{\text{TTI}}(\tau) + \left(\frac{1}{T}Q'(1) - \mu_\infty\right)[1 - C_{\text{TTI}}(\tau)] + T = -\frac{1}{T}\int_0^\tau Q'(C_{\text{TTI}}(\tau-\tau''))\frac{d}{d\tau''}C_{\text{TTI}}(\tau'')d\tau''\,. \tag{B.18}$$

An important limit that is used later on in the computation is when $\tau \to \infty$ and the variations of $C_{\text{TTI}}$ becomes irrelevant. This gives:

$$\mu_\infty = \frac{T}{1-q} + \frac{Q'(1) - Q'(q)}{T}\,. \tag{B.19}$$

### B.2.5 Regime 2: aging.

Starting from the evolution of the response function (B.13), in this regime the time derivative is negligible.

$$0 = -\mu_\infty\frac{\mathcal{R}(\lambda)}{t} + \clubsuit$$

with ♣ that can be separated into three terms $\clubsuit^{(1)}$, $\clubsuit^{(2)}$ and $\clubsuit^{(3)}$

$$\clubsuit = \int_{t'}^t R(t,t'')Q''(C(t,t''))R(t'',t')dt'' = \int_{t''\lesssim t} + \int_{t''\gtrsim t'} + \int_{\text{aging}} = \clubsuit^{(1)} + \clubsuit^{(2)} + \clubsuit^{(3)}\,.$$

In the first two integrals we can apply FDT respectively for $t''$ close to $t$ and for $t''$ close to $t'$:

$$\begin{aligned}
\clubsuit^{(1)} &= \int_0^\infty R(t,t-\tilde{t})Q''(C(t,t-\tilde{t}))\frac{\mathcal{R}(\lambda)}{t}d\tilde{t} = -\frac{\mathcal{R}(\lambda)}{t}\frac{1}{T}\int_0^\infty \frac{d}{d\tilde{t}}Q'(C_{\text{TTI}}(\tilde{t}))d\tilde{t} = \\
&= \frac{1}{T}\frac{\mathcal{R}(\lambda)}{t}\left(Q'(1) - Q'(q)\right),
\end{aligned}$$

$$\clubsuit^{(2)} = \int_0^\infty \frac{1}{t}\mathcal{R}(\lambda)Q''(q\mathcal{C}(\lambda))R(t'+\tilde{t},t')d\tilde{t} = -\frac{\mathcal{R}(\lambda)}{t}Q''(q\mathcal{C}(\lambda))\frac{1}{T}\int_0^\infty \frac{d}{d\tilde{t}}C_{\mathrm{TTI}}(\tilde{t})d\tilde{t} =$$

$$= \frac{1}{T}\frac{\mathcal{R}(\lambda)}{t}Q''(q\mathcal{C}(\lambda))(1-q)\,.$$

The last terms displays aging:

$$\clubsuit^{(3)} = \int_{t'}^t \frac{\mathcal{R}(\frac{t''}{t})}{t}\frac{\mathcal{R}(\frac{t'}{t''})}{t''}Q''\left(q\mathcal{C}\left(\frac{t''}{t}\right)\right)dt'' = \frac{1}{t}\int_\lambda^1 \frac{\mathcal{R}(\lambda'')}{\lambda''}\mathcal{R}\left(\frac{\lambda}{\lambda''}\right)Q''\left(q\mathcal{C}(\lambda'')\right)d\lambda''\,.$$

Combining these pieces together and using (B.19) we obtain an expression for the aging function of the response:

$$0 = \left[-\frac{T}{1-q} + \frac{Q''(q\mathcal{C}(\lambda))(1-q)}{T}\right]\mathcal{R}(\lambda) + \int_\lambda^1 \frac{\mathcal{R}(\lambda'')}{\lambda''}\mathcal{R}\left(\frac{\lambda}{\lambda''}\right)Q''\left(q\mathcal{C}(\lambda'')\right)d\lambda''\,. \quad \text{(B.20)}$$

Following the same steps in Eq. (B.12) and using again (B.19) we obtain the expression for the correlation:

$$0 = -\left[\frac{T}{1-q} + \frac{Q'(q\mathcal{C}(\lambda))(1-q)}{q\mathcal{C}(\lambda)\,T}\right]q\mathcal{C}(\lambda) + Q'(m_\infty)m_\infty +$$

$$+ \int_0^\lambda Q'(q\mathcal{C}(\lambda''))\mathcal{R}\left(\frac{\lambda''}{\lambda}\right)\frac{d\lambda''}{\lambda} + q\int_0^1 \mathcal{R}(\lambda'')Q''(q\mathcal{C}(\lambda''))\mathcal{C}\left[\left(\frac{\lambda''}{\lambda}\right)^{\mathrm{sign}(\lambda-\lambda'')}\right]d\lambda'' \quad \text{(B.21)}$$

**Remark 3 (generalized-FDT)** *In the derivation we never used generalized-FDT ansatz, $\mathcal{R}(\lambda) = \frac{1}{T}xq\frac{d\mathcal{C}(\lambda)}{d\lambda}$. A posteriori we can observe its validity as Eq. (B.20) and Eq. (B.21) collapse to a single equation as Eq. (B.21) is derived by $\lambda$ and generalized-FDT is used.*

### B.2.6  Characterisation of the unknown parameters.

In order to fully characterized the FDT Eq. (B.18) and the aging Eqs. (B.20-B.21), we need to determine the parameters $m_\infty$, $\mu_\infty$, $q$, the FDT violation index $x$ and $q_0 = q\mathcal{C}(0)$. We do not determine all of them, we consider only the few that are used in the analysis, but for sake of completeness we say how the five equations can be determined: Eq. (B.14) taking $t \to \infty$, Eq. (B.16) plugging the generalized FDT ansatz, Eq. (B.18) in the large $\tau$ limit, Eq. (B.20) in the limit $\lambda \to 1$, Eq. (B.21) in the limit $\lambda \to 0$.

In particular the $\lim \lambda \to 1$ of Eq. (B.20) gives

$$\frac{T^2}{(1-q)^2} = Q''(q)\,. \quad \text{(B.22)}$$

From Eq. (B.21), in the limit $\lambda \to 1$ and $\lambda \to 0$ we obtain

$$0 = \left[\frac{Tq}{1-q} - \frac{Q'(q)(1-q)}{T}\right] + Q'(m_\infty)m_\infty + \frac{x}{T}[q\,Q'(q) - q_0\,Q'(q_0)]\,, \quad \text{(B.23)}$$

$$0 = \left[\frac{Tq_0}{1-q} - \frac{Q'(q_0)(1-q)}{T}\right] + Q'(m_\infty)m_\infty + q_0\frac{x}{T}[Q'(q) - Q'(q_0)]\,. \quad \text{(B.24)}$$

In the regime where the system does not find a good overlap with the signal thus $m_\infty = 0$, the second equation gives the solution $q_0 = 0$. As $T$ tends to 0 (and $q$ tends to 1)

$$\frac{x}{T} = \frac{1}{q\,Q'(q)}\left[\frac{T}{1-q} - \frac{Q'(1)(1-q)}{T}\right] = \frac{1}{q\,Q'(q)}\left[\sqrt{Q''(q)} - \frac{Q'(1)}{\sqrt{Q''(q)}}\right]\,. \quad \text{(B.25)}$$

**Remark 4 ($\overline{R}$)** *In the large time limit, and using FDT, we have*

$$\overline{R} = \int_0^\infty R_{TTI}(\tau'')d\tau'' = -\frac{1}{T}\int_0^\infty C'_{TTI}(\tau'')d\tau'' = \frac{1-q}{T}\,, \quad \text{(B.26)}$$

*using Eq. (B.22) as $T \to 0$ we the result reported in the main text*

$$\overline{R} = \frac{1}{\sqrt{Q''(1)}}\,. \quad \text{(B.27)}$$

**Remark 5 (marginal states)** *Combining Eq. (B.22) with Eq. (B.19), we obtain:*

$$\mu_\infty = \sqrt{Q''(q)} + \frac{1}{T}\left[Q'(1) - Q'(q)\right] \tag{B.28}$$

*expanding $q \lessgtr 1$, and using again Eq. (B.22),*

$$\mu_\infty = 2\sqrt{Q''(1)}. \tag{B.29}$$

*As we explained in the main text, the distribution of the Hessian is associated to a semicircle of radius $2\sqrt{Q''(1)}$ and centred in $\mu$. This equation tells that asymptotically, if aging does not stops – as it happens if it jumps to the solution – the systems tends to the marginal states. We have shown a numerical confirmation of this property in Fig. B.1.*

**Remark 6 (threshold loss)** *As we show in the main text the Lagrange multiplier $\mu$ depends on the two losses as $\mu = -p\epsilon_p - 2\epsilon_2$ (or $\mu = T - p\epsilon_p - 2\epsilon_2$ for arbitrary temperature). Observing that the equation holds for any $\Delta_p$ and $\Delta_2$, in particular when they tend to infinity and therefore their contribution to the total loss becomes irrelevant, it follows from Eq. (11) (respectively Eq. (B.15)),*

$$\epsilon_p(t) = -\frac{1}{p}\left[Q_p'(m(t))m(t) + \int_0^t R(t,t'')\left[Q_p'(C(t,t'')) + Q_p''(C(t,t''))C(t,t'')\right]dt''\right] \tag{B.30}$$

*and analogously $\epsilon_2(t)$. We then write the expression for the total loss*

$$\epsilon(t) = -\frac{1}{p}Q_p'(m(t))m(t) - \frac{1}{2}\left[Q_2'(m(t))m(t)\right] +$$
$$+ \int_0^t R(t,t'')\left[Q'(C(t,t'')) + Q''(C(t,t''))C(t,t'')\right]dt''. \tag{B.31}$$

From the equation we established above and using the aging ansatz, one can obtain the asymptotic value of the loss for low signal-to-noise ratio, i.e. the loss of the minima trapping the dynamics. As anticipated in the footnote of the main text, we find that the asymptotic dynamical loss is not the one of the most numerous minima, $\epsilon_{th}^{KR}$, which can be obtained by the Kac-Rice method, but very slightly lower.

This is an interesting point. Recently, it was found in similar cases [11] that GF tends to marginally stable minima which are not the most numerous ones. However, for the purpose of this paper, in which the only important ingredient is that the minima trapping the dynamics are marginally stable, this is not relevant. Hence, our results on this aspect will be presented elsewhere.

**Remark 7 (threshold energy)** *The large time limit of Eq. (B.31) gives two threshold states. Applying the same scheme used in Eq. (B.16), i.e. integrating Eq. (B.30) for $t, t' \gg 1$ considering the two time-regimes gives*

$$\epsilon_{p,th}^{dyn} = -\frac{1}{p}\left[Q_p'(m_\infty)m_\infty + \frac{1}{T}\left[Q_p'(1) - q\,Q_p'(q)\right] + p\int_0^1 \mathcal{R}(\lambda)Q_p'(q\mathcal{C}(\lambda))d\lambda\right]. \tag{B.32}$$

*Applying the generalized fluctuation dissipation ansatz and Eq. (B.25) in the integral, and finally taking $T \to 0$ ($q \to 1$)*

$$\epsilon_{p,th}^{dyn} = -\frac{1}{p}\frac{Q_p'(1) + Q_p''(1)}{\sqrt{Q''(1)}} - \frac{x}{T}Q_p(1). \tag{B.33}$$

*The threshold energy will be given by the some of two contributions, giving*

$$\epsilon_{th}^{dyn} = -\frac{Q'(1)}{\sqrt{Q''(1)}} - \frac{Q(1)\left(Q''(1) - Q'(1)\right)}{\sqrt{Q''(1)}Q'(1)}. \tag{B.34}$$

- 

## C   Numerical Simulations of Gradient-Flow

Figure C.1: Evolution of the loss in time from numerical simulations realised over 100 instances of disorder and noise, for the spiked matrix-tensor model with $p = 3$. The simulations has been done with systems of size $N = 2^{16} - 1 = 65535$ with parameters $1/\Delta_2 \in \{1.5, 1.9, 2.3, 2.7\}$ and $\Delta_p = 1.0$.

In order to evaluate Gradient flow dynamics we discretized time and evaluated Eq. (4) numerically using effectively gradient descent

$$\sigma_i^{t+1} = -\mu^t \sigma_i^t - \frac{\partial \ell(\boldsymbol{\sigma}^t | \boldsymbol{T}, \boldsymbol{Y})}{\partial \sigma_i^t} \,. \tag{C.1}$$

In our experiments we run the dynamics on numerous realisations of the problem for different values of the parameters at $p = 3$. Given a signal $\boldsymbol{\sigma}^* \in \mathbb{S}^{N-1}$, the number of computations per interaction scales as $N^3$, which makes the system hard to simulate for large $N$. In order to increase the size of the system, we considered a diluted system, as proposed in [12], instead of the real system, such that the first and the second moment of the loss, in the leading order on $N$. In the original system the (hyper)-graph of interaction is fully connected and counts $N^3/3!$ (hyper)-edges for the tensor and $N^2/2$ edges for the matrix. In the diluted systems we replace the (hyper)-graphs by graphs less connected in particular we take $N^2$ (hyper)-edges for the tensor and $N\sqrt{N}$ edges for the matrix. In systems with spherical variables there is a known problem [13] associated with reducing too much the number of interaction. In general given a tensor of order $p$ if the number of interaction becomes less then $N^{p-1}$ the system tends to favour a finite number of (hyper)-edges and aligns completely with them. The dynamics then converges to a final configuration where $O(p)$ spins have value of order $O(1)$ and the rest is of order $O(1/\sqrt{N})$. In order to have the same averages for the observables — such as overlap with the signal and loss — called $\#(\cdot)$ that counts the number of interactions, we multiplied the variances of the noise by $N^{3/2}/(3!\#(\boldsymbol{T}))$ and $N/(2\#(\boldsymbol{Y}))$ respectively the tensor noise and the matrix noise.

Using this observation in the code we obtain a simple algorithm that given a $dt$ approximate gradient flow by computing a gradient descent dynamics, with $dt = 1.0$ in the simulations. This value was chosen observing that in the runs the algorithm always descends in terms of loss and not appreciable difference appeared reducing it further. The code is made available and attached to this paper. Using this code we were able to simulate systems of the size $N = 2^{16} - 1 = 65535$ and reduce finite size effects. Fig. C.1 shows the average over different initialisation and realisation of the noise for the parameters presented in the paper $\Delta_p = 1.0$ and $1/\Delta_2 \in \{1.5, , 1.9, , 2.3, , 2.7, \}$. In the figure we use a continuous line surrounded by a shadow to represent mean and standard deviation under a Gaussian hypothesis, individual simulations are represented using dashed-dotted lines. For $p = 3$ and $\Delta_p = 1.0$ the critical threshold for gradient flow occurs at $1/\Delta_2^{\text{GF}} = 2.0$ and in fact we observe

that the green line ($1/\Delta_2 = 1.9$) shows finite size effects and some simulations find good overlap with the ground truth. To conclude the figure shows a very good agreement with the averaged value evaluated using CHSCK equations, see Fig. 4-b. In particular is evident how all the dynamics tends to the threshold states, whose corresponding losses are drown with horizontal dotted lines, before eventually find the good direction and then the signal.