[Reviews · NeurIPS 2019]

Reviewer 1



ORIGINALITY, QUALITY, CLARITY, and SIGNIFICANCE Originality: I view the arguments and ideas (especially using Kac-Rice + BPP to understand instability of non-convex losses) to be a nice idea, which to my knowledge is new. Quality: The theoretical arguments provided by the authors are sound and the numerics are convincing. Clarity: The paper is clearly written, except for a few points I mention below. Significance: While the author's idea are developed directly in the context of tensor-matrix models, I believe the analytical ideas and proposed mechanism for the formation of saddles that are useful for optimization in non-convex landscapes may be important for many other models. DESCRIPTION OF MAIN RESULTS The spiked tensor-matrix model seeks to recover a ground truth signal \sigma_* (assumed to be not the sphere in N dimensions) from a superposition of Gaussian perturbations of the rank one matrix and higher rank one tensors built from \sigma_*. The loss function has two parameters \Delta_p and \Delta_2 that together determine the signal to noise ratio, which is inversely proportional to \Delta_2. The main result, from my point of view, in this article is an analytical formula for a critical threshold \Delta_2^{GF} (equation (5)) so that only for \Delta_2 below this threshold does gradient flow find a minimum that is correlated with the signal. This result holds in the large N limit for \Delta_p fixed. The key point of this result is that \Delta_2^{GF} is strictly above what the authors call \Delta_2^{triv}. The latter threshold is one that governs when the entire loss landscape has no spurious local minima. Thus, the most exciting result for me is the proposal for a mechanism for why, even in the theoretical presence of many spurious local minima gradient descent typically finds a solution that is positively correlated with the underlying signal. The basic idea is in retrospect simple and, as the authors point out, may well be play a part in analyzing a range of other models. The authors give two explanations, but I feel unqualified to comment on the one via the CHCKR equations from spin glass theory. The other explanation analyzes the effect the of \Delta_2 on the form of the Hessian at critical points. This is related to Kac-Rice approaches to the complexity of spin glasses and was obtained before. Specifically, the Hessian at a critical point is the sum of a GOE matrix, a multiple t(\ep) * I of the identity and rank one perturbation with strength \theta(\Delta_2, \Delta_p). Here \ep is the total energy of the critical point in the uncorrelated signal regime in which \sigma is uncorrelated form the ground truth (as it is at the start of training). The key point is that the left edge of the spectrum for the Hessian will touch 0 precisely when t = 2. Since t is a monotone function of \ep, this gives a critical energy threshold, which the authors denote by \ep_{th}^{KR} for when the rank one perturbation has a chance of producing a negative eigenvalue. The BBP transition says that when \theta = 1 a single eigenvalue appears on the left edge of the spectrum. Above the energy \ep_{th}^{KR}, critical points will be saddles and thus one expects gradient flow to approach a critical point of energy \ep_{th}^{KR}. At this energy, when \theta at least one, this will also be a saddle from which a downward flow direction will correlate the signal with \sigma_*. COMMENTS Overall, I found the article to be very interesting. Connecting Kac-Rice to BPP is a nice idea. For the most part the article is easy to read, save perhaps for a few relatively minor points that I hope the authors might address in a final version: (1) It was confusing for me that \Delta_p, \Delta_2 are variances of the \eta’s since I would have thought it is strange to divide a Gaussian by its variance. Is that really the case or are \Delta_p, \Delta_2 standard deviations? (2) On line 66 “correlated the large spike” should be “correlated with the large spike” (3) On line 67, there is a space missing after “large N” (4) Around lines 96-98, the writing makes it seem like the Kac-Rice formula was first developed in the cited papers. The authors I think mean that the KR formula was applied in the spin glass world in those papers. The KR formula itself goes back to the late 30’s and 40’s. (5) I found the discussion on lines 101-107 confusing. If you are replacing the complexity by its averaged (annealed or quenched) versions, then some discussion of the fluctuations of this random variable seems warranted. Are you claiming the variance is small? (6) I found the conclusion on line 145 that \ep = \ep_{th}^{KR} are the “most numerous and the marginally stable ones” to be confusing. I tried in the summary of results above to give what I think is the argument you had in mind. In any case, I think it would be useful to a reader if it were spelled out a bit more. (7) On lines 159-161, when you refer to the unstable direction, it is not clear form the formula for H that the e_1 direction is in fact the direction of \sigma_*. I think you mean that, directly from the KR analysis, e_1 = \sigma_*. If that’s right, it might be helpful to say explicitly. (8) On line 225, “identicall” has an extra ell. UPDATE AFTER REBUTTAL I thank the authors for their careful response. I've slightly revised my score for this submission (from a 9 to an 8) since it was brought to my attention that a bunch of the previous analysis of both Kac-Rice and CHCKR in the matrix-tensor setting was done in previous papers. I hadn't quite understood this when I wrote my original review. Still like the paper very much, and I hope it will be accepted.

Reviewer 2



Originality: The work contains new analytic results for the spiked matrix-tensor model (in a limit in which they specify -- e.g. high-dimensional signal vector to be recovered) governing the transition to a "gradient-flow easy" regime. The analytic results are in excellent agreement with the numerics. Analysis of the Hessian at a certain subset of stationary points reveals the mechanism by which gradient-flow is able to find good solutions in this intermediate regime (before landscape trivialization), despite the presence of spurious local minima. Quality: The work appears to be of high quality. The authors consistently specify which theoretical techniques and results are rigorous and which are based on physically reasonable guesses. Additionally, results which have appeared in prior work are carefully cited. The agreement between the theory and numerics is excellent, and the authors use multiple approaches to check that their results and intuitions are consistent. Clarity: The text and figures are easy to understand. Significance: The analytic results and resulting intuitive picture for this particular model is of interest to theorists working on high-dimensional inference. The intuition, and possibly some theoretical aspects, may carry over or be generalizable to other settings such as neural networks. There are relatively few semi-rigorous results on the behavior of gradient flow in a nonconvex optimization landscape, so this work appears to make some notable contributions.

Reviewer 3



The reviewer has the following concerns: 1. Though the phenomenon of avoiding meeting exponentially many spurious local minimizers is interesting, running gradient flow dynamics from random initialization on this model seems artificial. A spectral initialization of the spiked matrix can already give an estimates that has non-trivial correlations when $\Delta_2$ is smaller than $1$. Why should one consider gradient flow from random initialization rather than gradient flow from spectral initialization? 2. In terms of technical contributions, early works already calculated the complexity and CHSCK formula, where the landscape and AMP threshold was given. The numerical computation of GF threshold was also given. The formula of GF threshold in this work was extracted from algebraic analysis of the complexity formula and CHSCK formula. This is an interesting analysis. However, it is hard to say if this contribution is enough for NeurIPS. ------- After reading the authors' response and discussion with other reviewer, I increased my score to 6. My concern was that, most of the hard works are already done in the previous papers, and the matrix-tensor model may not be interesting enough. Now I think the heuristic Hessian analysis and perturbation argument for CHSCK in this paper is interesting. Moreover, since the matrix-tensor model is accepted by the community as an example to study computational complexity, it is good for people to have a better understanding about this model.

[Author Response · NeurIPS 2019]

We are grateful for the detailed reviews and comments that will help us to make our paper clearer.

**Referee #1**

We thank for remarks (2), (3), (8) that we will fix.

Remark 1: $\Delta_2$ and $\Delta_p$ are the variances, not standard deviations. The loss comes from terms such as $\exp[-(Y_{ij} - \sqrt{N}\sigma_i\sigma_j)^2/(2\Delta_2)]$ where the cross-term appearing in the loss is $Y_{ij}\sqrt{N}\sigma_i\sigma_j/\Delta_2$.

Remark 4: We thank the referee for pointing our imprecision. We will add the references to the original work: "S. O. Rice, Mathematical analysis of random noise, Bell System Tech. J., 24 (1945), 46–156"; "M. Kac, On the average number of real roots of a random algebraic equation (1943)" summarized very nicely in "Adler, Robert J., and Jonathan E. Taylor. Random fields and geometry. Springer 2009". The references we gave originally is the application of the Kac-Rice method in the high-dimensional setting.

Remark 5: Bounding the variance would be another way to justify that the annealed calculation is tight. We have not done that, but will look into it. Having the quenched complexity equal to the annealed complexity is only asymptotic and in density. It implies that the annealed complexity bound is tight, but it does not itself imply a very strong bound on the variance. We agree with the referee, that this passage should be clarified to explain better why this is important and why this means that the annealed complexity bound is actually tight. We will adjust the final version.

Remark 6: The reasoning of the referee is indeed what we had in mind. We will rephrase the argument in the final version following the suggestion of the reviewer.

Remark 7: We thank the reviewer for the remark we will clarify in the final version. In the formula we select a specific reference frame where we can easily represent the Hessian. In this frame the basis element $e_1$ is the vector aligned with the part of the signal that is tangent to the estimator. Calling $\boldsymbol{\sigma}^*$ the signal and $\boldsymbol{\sigma}$ the estimator, and saying that the estimator is an element in the basis, we have the relation: $\boldsymbol{\sigma}^* = \sqrt{1-m^2}\,\boldsymbol{e}_1 + m\,\boldsymbol{\sigma}$ with $m = \langle\boldsymbol{\sigma},\boldsymbol{\sigma}^*\rangle \in [-1,+1]$.

**Referee #3**

We summarise the state-of-the-art for the model and the technical contributions:

The literature on this specific model is so far limited. As far as we know it was introduced in [21], and studied in [20].

The main theoretical techniques the CHSCK equations and the (annealed = log of expectation) Kac-Rice method were used in [20]. In this paper we derive the replica symmetric quenched (= expectation of the log) Kac-Rice equations justifying (non-rigorously) that the annealed bound is tight in the context we use it, this is one technically involved contribution. Another technically involved contribution (non-rigorous, but conjectured exact) is the analysis of the large time solution of the CHSCK equations that has not been done in existing literature, it has only been studied in related optimization problems, not in inference problems where the spike is to be recovered. Finally the interpretation of the Kac-Rice calculation in terms of the relation of the effective Hessian and the behaviour of the dynamics is also original. For a summary of the main non-technically-involved but in our opinion broadly interesting contributions see the answer to Referee #4.

**Referee #4**

Remark 1: We consider gradient flow from random initialization because among the algorithms we are able to analyze this is closest to what is done in practice for training neural networks. Our point is not to search for the best algorithm for this specific model – we believe that would be the AMP algorithm (in agreement with conjectures of previous works). Our main interest is to understand the trajectory Gradient Flow (GF) takes in the non-convex high-dimensional landscape. While understanding the behaviour of variants of GF is also a very interesting question, we start with the most basic algorithm that has non-trivial behaviour.

Remark 2: Referee's assessment of what was known from previous work, notably ref. [20], is correct. We summarised the technical contributions of the present work with respect to existing literature in the answer to Referee #3. Additionally, the reason why we are persuaded the present paper is significant is, as Referee #1 puts it "The basic idea is in retrospect simple and may well play a part in analysing a range of other models." Indeed, our reasoning leading to the formula for $\Delta_2^{\mathrm{GF}}$ is in retrospect not restricted to the present model, and ends up way simpler than the full CHSCK analysis or the full Kac-Rice complexity calculation. The mechanism of converging to the threshold states and then escaping from them can also be tested numerically even in models that are not amenable to analytic description. This makes the results of the present work widely testable and applicable to other settings than the present model. We shall highlight these in the final version.

[Meta-Review · NeurIPS 2019]

This paper analyzes the critical points in the spiked matrix-tensor model, framing it as a proxy for more general models in high-dimensional inference. It examines the signal to noise ratio, its effect on spurious local minima, and the behavior of gradient flow. I agree with the reviewers' consensus that this non-trivial analysis would be interesting to the NeurIPS community and I recommend acceptance.